# Impact of Zinc Transport Mechanisms on Embryonic and Brain Development

**DOI:** 10.3390/nu14122526

**Published:** 2022-06-17

**Authors:** Jeremy Willekens, Loren W. Runnels

**Affiliations:** Department of Pharmacology, Rutgers-Robert Wood Johnson Medical School, Piscataway, NJ 08854, USA; jeremy.willekens@rutgers.edu

**Keywords:** zinc, zinc deficiency, development, brain, fetal programming, TRPM7

## Abstract

The trace element zinc (Zn) binds to over ten percent of proteins in eukaryotic cells. Zn flexible chemistry allows it to regulate the activity of hundreds of enzymes and influence scores of metabolic processes in cells throughout the body. Deficiency of Zn in humans has a profound effect on development and in adults later in life, particularly in the brain, where Zn deficiency is linked to several neurological disorders. In this review, we will summarize the importance of Zn during development through a description of the outcomes of both genetic and early dietary Zn deficiency, focusing on the pathological consequences on the whole body and brain. The epidemiology and the symptomology of Zn deficiency in humans will be described, including the most studied inherited Zn deficiency disease, *Acrodermatitis enteropathica*. In addition, we will give an overview of the different forms and animal models of Zn deficiency, as well as the 24 Zn transporters, distributed into two families: the ZIPs and the ZnTs, which control the balance of Zn throughout the body. Lastly, we will describe the TRPM7 ion channel, which was recently shown to contribute to intestinal Zn absorption and has its own significant impact on early embryonic development.

## 1. Introduction

Zinc (Zn) is the second most abundant transition metal in the human body [1]. Unlike iron (Fe) and copper (Cu), Zn is redox neutral and is reactive as a Lewis acid in biological reactions [2]. In cells, the metal is maintained in the Zn state and shows flexible and dynamic coordination geometry with nitrogen, oxygen sulfur ligands integrated into histidine, glutamate, aspartate, and cysteine residues in proteins [3]. This feature allows for rapid shifts of conformations in proteins, which nature has found useful, employing Zn as a cofactor in over 300 enzymes spanning all enzyme classes [4]. During enzymatic reactions, Zn may have either a direct catalytic role or an essential role in stabilizing the structure of catalytic enzymes. Thus, Zn is used as a structural element in over 3000 proteins [4], including growth factors, cytokines, receptors, enzymes, and transcription factors. Indeed, it has been estimated that approximately ten percent of the proteome is Zn-containing proteins, underscoring the critical role Zn plays in biological systems [5]. Many of these proteins contain Zn finger motives, finger-like structures that stabilize the conformation of many proteins, including nuclear hormone receptors that bind to estrogens, thyroid hormones, vitamin D, and vitamin A, allowing the steroid receptors to bind to DNA. Underscoring the importance of Zn to this class of proteins, metalloproteomic analysis has revealed that 44% of Zn proteins regulate DNA transcription; more than 2000 transcription factors that work to regulate gene expression require Zn to maintain their structural integrity and bind to DNA [4]. In contrast, only a small percentage of Zn proteins are involved in cell signaling (5%), although they have important cellular functions. For example, Zn is critical to the activity of Matrix Metallopeptidases (MMPs) [6], the oxidative stress response through Superoxide Dismutase 1 & 3 (SOD1 & SOD3) [7], maturation of insulin precursor through Insulin-Degrading Enzyme (IDE) [8], and global methylation process via Methionine Synthase (MS, *Mtr*) stabilization [9]. Thus, Zn as a metal has impact on a multitude of biological processes within cells.

The total Zn concentration in eukaryotic cells is relatively high (0.1–1 mM), but the actual amount of free Zn ranges from ~pM to ~nM values, depending on the cell type [10]. Analysis of the distribution of Zn in cells found that 54% are in the cytoplasm, 30–40% are in the nucleus, and the remaining 10% in the membrane (10%) [11]. Zn can be found in intracellular organelles, including lysosomes, mitochondria, the endoplasmic reticulum (ER), and the Golgi apparatus [12]. Much of the metal (54%) in cells is non-exchangeable and non-reactive immobile Zn that is tightly bound to predominantly proteins [10]. In addition to the immobile Zn there is also “mobile reactive Zn” that is loosely bound to a ligand and is an exchangeable reactive pool of Zn (44.7%). The remainder Zn in cells is in the form of a reactive pool of the metal that is unbound (free Zn), with a concentration in the picomolar range [10]. The availability of Zn inside of the cell has significant effects on many proteins’ functions, and changes in the concentration of free Zn can modulate signaling cascades. Not surprisingly, given the number of proteins that bind to Zn and employ it as a metal cofactor, cellular deficiency of Zn can be detrimental to cells. However, too high a level of free Zn can also be damaging [13], in some instances causing irreversible protein aggregation and cellular dysfunction, particularly in Alzheimer’s disease (AD) [14]. As we will discuss below, organismal deficiency of Zn also produces profound defects during human development and later in adult life. Thus, the Zn homeostasis within cells and the body must be tightly regulated by a multitude of transporters and homeostatically maintained by absorption of the trace mineral.

## 2. Sources, Intakes, and Absorption Cycle of Zinc

Next to Fe, Zn is the second most abundant major trace mineral in the human body, with bodily stores for a 70 kg male totaling 2.3 g [15]. Zn is present in all the tissues of the human body, with the highest cellular concentrations found in skeletal muscles (60%), bones (30%), and liver (5%) [16]. Serum Zn accounts for only ~0.1% of the body’s Zn, with 80% loosely bound to albumin and 20% tightly bound to α2-macroglobulin (Figure 1). Zn is mainly transported and delivered to peripheral tissues by albumin, constituting the exchangeable pool of Zn, and by α2-macroglobulin, which is the non-exchangeable reserve of Zn [17]. To maintain bodily stores of Zn, approximately 0.1% of the body Zn is replenished daily through diet. According to the Dietary Reference Intakes (DRIs) developed by the Food and Nutrition Board (FNB) at the Institute of Medicine of the National Academies [18] daily intake recommendations for Zn are 11 mg for men and 8 mg for women (Table 1 [19,20]). These values can be increased when the demand is higher, for example during pregnancy. The vast majority of Zn intake comes from food (Table 2). Oysters contain more Zn than any other aliment; however, red meat and poultry are the most common sources of Zn in regular diets. Finally, Zn is also found in beans and fortified breakfast cereals. More than 30 proteins, including ZnT (Zinc transporter protein) and ZIP (Zinc/Iron-regulated transporter-like Proteins) transporters, as well as the TRPM7 ion channel, are involved in the maintenance of systemic and cellular Zn homeostasis in mammals. Zn is absorbed throughout the whole small intestine [21]. The highest rate of absorption occurs in the jejunum; however, the largest amount of Zn is absorbed in the duodenum, because the highest Zn concentrations after a meal occur there. Zn uptake in the small intestine is achieved by two mechanisms, a saturable, carrier-mediated process and a non-mediated, passive process [22]. As we will discuss later in this review, a mutation of the gene coding for ZIP4 (*SLC39A4*), is responsible for the Zn malabsorption disorder *Acrodermatitis enteropathica* (*AE*) [23]. In addition, a more recent study, has also highlighted a role for the TRPM7 channel in intestinal Zn absorption [24].

Within enterocytes, Zn can be found in three different states: free in the cytosol, bound to metallothioneins [25], or stored within vesicles [26]. After Zn is absorbed into the enterocyte, two proteins, metallothionein (MT) and the transporter ZnT7, are primarily responsible for the movement of Zn through the cell [27]. Finally, the ZnT1 transporter facilitates Zn transfer from the enterocyte into the circulation. The body’s ability to absorb Zn increases up to 90% when the availability of Zn is limited [22]. When too much Zn is taken in, Zn is secreted into the lumen of the small intestine, where it can also be reabsorbed to maintain homeostasis or excreted into the feces [28] (Figure 1). Kidney secretion of Zn into urine remains constant and is not considered as a way of regulation of Zn homeostasis, except under extreme conditions, such as severe Zn deprivation [29]. 

## 3. Epidemiology of Zinc Deficiency

About 17% of the global world population are at risk of Zn deficiency, with the vast majority residing in Africa and Asia [30]. In fact, Zn deficiency has become so prevalent that it has been recognized as a major worldwide health issue. Worldwide, Zn deficiency has been estimated to cause more than 450,000 deaths in children under 5 years [31], contributing to 800,000 total deaths annually. Furthermore, Zn deficiency is responsible for about 20% of perinatal mortality [32]. Malnutrition, and more specifically undernutrition, is the main cause of Zn deficiency in least developed countries, whereas developed countries can suffer from Zn deficiency because of inadequate diets [33]. However, as we will discuss below, genetic alterations in Zn metabolism pathways and homeostasis can also be the cause of the same symptoms produced by insufficient Zn intake [34].

## 4. Impact of Zinc on Health, Disease, and Development

### 4.1. Clinical Presentation of Zn Deficiency

In 1961, when Zn was first identified as an essential micronutrient for humans, it was recognized that Zn deficiency can produce a broad array of symptoms [35]. The most common symptoms and first signs of Zn deficiency are cutaneous lesions and diarrhea. More severe symptoms include severe anemia, growth retardation, hypogonadism, skin abnormalities, and mental lethargy. Since then, a growing array of symptoms attributed to nutritional Zn deficiency has been described, including persistent diarrhea, alopecia, taste disorders, immune insufficiency, brain dysfunctions, impairment of wound healing, loss of appetite, chronic inflammation, and liver disease (for detailed reviews, see [29,32,35,36,37,38,39]). In addition, Zn deficiency in males can affect spermatogenesis as well as global sperm quality, potentially leading to infertility [40,41]. Zn deficiency can cause also contribute to male infertility through hypogonadism, inflammation, antioxidant depletion, or disruption of testosterone and progesterone production [42]. Zn deficiency is also associated neuropsychological changes, such as emotional instability, irritability, and depression, as observed both in mouse [43] and in human [44]. This is in part due to the fact that free Zn within vesicles plays an essential role in the brain and more specifically in the modulation of synaptic function [45]. Although the importance of Zn was first highlighted through medical cases of Zn deficiency and its consecutive symptoms, such as alteration of immune response or neurological disorders [34], dysregulations of Zn homeostasis can be the starting point of numerous pathologies, including arterial hypertension [46], metabolic syndrome [47], neurodegenerative diseases [48,49], and cancer [50]. Therefore, dysregulation of Zn homeostasis will have complex consequences at physiological and cellular levels, contributing to many pathologies, including neurological disorders [51].

Dietary Zn deficiency has a high prevalence in infants under the age of 5 in less developed countries and can then be confused with transient infantile Zn deficiency (TIZD), a pathology causing Zn deficiency that affects newborns under 6 months that are exclusively breastfed. The TNZD (transient neonatal zinc deficiency) acronym can also be used in some publications to refer at the same pathology. In humans, numerous genetic variants in the gene encoding ZnT2 (*SLC30A2*) are responsible for TIZD, which is caused by defective secretion of Zn in mothers’ breast milk [52]. Infants suffering from TIZD display Zn deficiency due to low milk Zn concentration in their nursing mothers, which cannot be corrected by maternal Zn supplementation. The symptoms of TIZD cannot be distinguished from AE, the genetic form of Zn deficiency, except by the fact that Zn deficiency is detected after weaning in AE, since breast milk displays protective effects by bringing maternal source of Zn [53]. Mutations in ZIP4 mutations are known to cause AE, which is a rare autosomal recessively inherited disorder of intestinal Zn malabsorption that is also called “genetic Zn deficiency,” [23,54,55,56]. *AE*, the most known and studied genetic form of Zn deficiency, is an autosomal recessive disease affecting 1:500,000 newborns, causing a defect in Zn absorption [57]. In contrast to *AE*, TIZD is reversible and an oral Zn supplementation can rapidly improve the symptoms [58]. However, despite the efficiency of supplementation, Zn deficiency remains a major concern because of its irreversible effects during embryogenesis. This is due to the fact that during early life, the needs for Zn are higher, and a severe dietary deficiency during the first six months of life can have deleterious developmental consequences. Indeed, Zn deficiency can dramatically affect every step of development [59], beginning with fertilization.

### 4.2. Impact of Zn Deficiency on Early Embryonic Development

Zn has a preponderant role in cellular homeostasis and as described above, Zn deficiency has a high prevalence around the world. Thus, Zn deficiency has dramatic consequences at every step during development. In fact, Zn deficiency adverse consequences on the offspring start before fertilization. Much of this review necessarily focuses on the maternal aspect of Zn deficiency, but growing evidence demonstrates the importance of adequate nutrient intake for spermatogenesis, as well as global sperm quality and fertility [40,41]. Indeed, Zn deficiency has been found to impair spermatogenesis and can cause male infertility through hypogonadism, inflammation, antioxidant depletion, or disruption of testosterone and progesterone production [42]. However, the molecular mechanisms behind these alterations are not well understood and remain understudied. Some articles nonetheless focus on the consequence of both maternal and paternal Zn deficiency on Zebrafish embryogenesis [60,61], but the effects of paternal Zn deficiency cannot be distinguished from the maternal effects. What is clear, however, is that Zn deficiency has important consequences on male infertility.

Recent studies have tried to elucidate the molecular mechanisms behind male infertility. For example, Bian and colleagues used a mouse model to show that a low Zn diet decreases Zn within testis tissues, causing testis histopathological abnormalities, especially at the level of seminiferous tubes. These changes are associated with dysregulations of FAK (Focal Adhesion Kinase) and TGF-β1 (Transforming Growth Factor Beta 1) expression, suggesting potential molecular keys to better understand Zn deficiency-induced spermatogenesis disorders [62]. Moreover, Kothari and colleagues showed that seminal plasma Zn in men between 21 and 50 years was significantly lower in abnormal ejaculates than in normal ejaculates [63]. Furthermore, they also highlighted a correlation between seminal plasma Zn and the different seminogram parameters, which implies that low seminal plasma Zn might be a cause of impaired sperm functions. Similar results have been found in various animal models, such as goat [64], poultry [65] and rat [66]. Therefore, Zn supplementation maybe a useful approach to rescue these detrimental phenotypes, as showed by several studies. Indeed, some causes of male infertility, such as asthenozoospermia (a sperm abnormality characterized by a decreased motility of spermatozoa), can be reduced by Zn supplementation, through prevention of oxidative stress, and by a reduction in apoptosis [67]. While paternal Zn deficiency is not highly relevant to the impact of Zn deprivation on offspring development, it remains a pertinent marker in global populations in order to better understand causes and molecular consequences of male infertility. Maternal Zn deficiency, on the other hand, can have profound impact on development, beginning soon after conception.

Upon oocyte activation, mammalian eggs release billions of Zn ions in an exocytotic event termed the “Zn spark” [68]. A retrospective analysis of Zn spark profiles revealed that parthenotes and zygotes that developed into blastocysts released more Zn than those that failed to develop. Thus, a robust Zn spark profile appears to be associated with embryo quality. Zn deficiency also affects development of the embryo. Female rats fed a Zn-deficient diet during pregnancy have smaller litters, with pups displaying growth retardation and multiple developmental abnormalities affecting the lungs, heart, urogenital system, and brain [69]. This is consistent with what is observed in humans, as maternal Zn deficiency during pregnancy in humans increases the risk of fetal growth restriction [70], with a clear association between maternal Zn level during pregnancy and the risks of low birth weight and small gestation age infants (Figure 2). Additional studies on human populations have confirmed the effect of maternal Zn deficiency on growth retardation [71,72], but have also uncovered congenital abnormalities [73], immune system failures [74,75], reproductive defects [76,77], and more generally an increased susceptibility to complex pathologies [46,47]. Unfortunately, studies conducted on humans are intrinsically descriptive and those focusing on molecular impact of Zn on development are rare. However, much has been learned from studies in animals on the effect of Zn on growth and development.

More recent studies on mice have demonstrated additional developmental defects (Figure 2). Thus, females fed a Zn-deficient diet for 4–5 days before ovulation show a higher incidence of embryos having decreased lengths and weights, with approximately half of the embryos exhibiting delayed or aberrant neural tube development [78]. Interestingly, embryos cultured in a medium containing the Zn chelator TPEN (N,N,N′,N′-tetrakis(2-pyridinylmethyl)-1,2-ethanediamine) demonstrated that Zn deficiency can cause neural tube closure defects through an increase in p53-dependent apoptosis [79,80]. Preconceptional Zn deficiency also interfered with trophoblast cell differentiation and implantation and caused a high incidence of pregnancy loss [78]. When mouse blastocytes are cultured in a low Zn medium, there is a reduction in extraembryonic endoderm due to an increase in apoptosis [81]. Embryonic stem cells (ESCs), which are the chief cellular source during embryogenesis in multicellular organisms, give rise to the three embryonic germ layers (mesoderm, endoderm, and ectoderm). A recent study found that Zn supplementation helps to sustain a stable pluripotent phenotype [82] Whether a deficiency in Zn affects the population of stem cells in the developing embryo has not been studied.

### 4.3. Origin of Late Developmental Effects

According to the fetal programming hypothesis, detrimental events taking place during development can have dramatic consequences in adulthood and especially increase susceptibility to complex pathologies [47,83,84] (Figure 2). Among others, these detrimental events include, for example, exposure to endocrine disruptors [85], alcohol [86], or malnutrition [87]. Because of the importance of Zn at cellular level, Zn deprivation during embryogenesis is expected to have dramatic effects on fetal development, as well as give rise to pathologies that present later in adulthood. For instance, the epigenome of the oocyte is dramatically remodeled during oogenesis. As the oocyte nears ovulation, major changes in chromatin structure and biochemistry occur to prepare the oocyte for fertilization and embryonic development. Indeed, oocyte chromatin methylation as well as preimplantation development is remarkably disrupted in female mice fed Zn-deficient diets for 4–5 days before ovulation [78]. More specifically, the authors found a dramatic decrease in histone H3K4 trimethylation and global DNA methylation in Zn-deficient oocytes. In addition, a three to twenty fold increase in the transcript abundance of repetitive elements (*Iap*, *Line1*, *Sineb1*, *Sineb2*) was observed, accompanied by an aberrant decrease in oocytes specific *Gdf9*, *Zp3* and *Figla* mRNAs. A recent study has shown that a maternal Zn deficiency during development could increase the susceptibility to development of breast cancer in a female mouse model [88]. Molecular and epigenetic dysregulation were detected in the adult mammary gland of the offspring, underscoring the potential long-term pathological consequences of gestational Zn deficiency. Furthermore, it has been shown that a prenatal, as well as a postnatal, Zn deficiency can induce cardiac alterations through apoptosis and inflammation in rat hearts [89]. While these effects are less dramatic in females and can mainly be reverted in both sexes by a Zn-adequate diet, they can still increase susceptibility to cardiovascular diseases if Zn deficiency is not diagnosed. However, among all the organs, it is the brain that is particularly vulnerable to nutrients or vitamin deprivation during development, producing deleterious early events can ultimately lead to dysregulations of molecular pathways that produce structural abnormalities in addition to cellular and molecular alterations [90,91,92].

### 4.4. Long-Term Consequences of Zn Deficiency on Brain Function during Development

Zn deficiency has a high prevalence in less developed countries and, as described above, has adverse consequences during early development. Additionally, Zn deficiency can contribute or give rise to various pathologies after embryogenesis is complete. In fact, a high proportion of these complex pathologies have a developmental origin. This is the “fetal programming hypothesis,” in which detrimental events that occur during development cause nonreversible, long-term changes that increase the susceptibility to complex pathologies after birth, during childhood, and in adulthood (Figure 2). Above, we described the global malformations that occur first during embryogenesis, which produce birth defects as a result of gestational Zn deficiency. However, some organs are extremely sensitive to the disruption of their homeostasis during the developmental period, and the consequences of these alterations will only be detected after birth, when pathological symptoms will appear. Thus, many neurological and behavioral disorders have been shown to have a developmental component. This is for example the case for schizophrenia, which has a genetic component with risk variants and risk loci associated with higher incidence of the pathology [93], but also fetal origins, including in utero infections [94] gestational starvation [95,96], and smoking [97]. Similar evidence also exists for autism spectrum disorder (ASD) [98,99,100], as well as anxiety and depression-related behaviors [101,102,103,104]. Concerning Zn, a recent study showed that gestational Zn deficiency produced disturbances in microbiota composition that caused neuroinflammation in the hippocampus, increasing the risk for depression later in adult life [105]. It also has been suggested that metal ions, especially Zn, could be linked to pathophysiology of ASD [106], especially since Zn deficiency is highly represented in autistic children under the age of 3 [107].

Structural alterations in response to early Zn deficiency have been shown to occur in different brain regions, including the hippocampus and the cerebellum. It was found that the granular layer of cerebellum has a lower density of neurons under Zn deprivation, and these neurons have shorter and less branched dendrites compared to control rats [108]. In addition, Zn deficiency can affect multiple cell types in the brain. Indeed, an alteration of astrogliogenesis was observed in the cortex of pups subjected to maternal Zn deficiency [109]. The decreased number of astrocytes in the brain cortex of offspring extends into adulthood, highlighting the long-term impact of early Zn deprivation.

Changes to the brain can have far-ranging consequences (Figure 2). For example, it has been shown that severe Zn deficiency in monkeys before weaning causes cognitive deficits that can be prolonged during childhood and then lead to long-term learning and memory alterations [110]. This has also been observed in rat, where early Zn deficiency during pregnancy and lactation leads to defects in learning and short-term memory, as well as motor activity alterations of the pups [111,112]. These alterations have been linked with increased apoptosis and shrinkage of hippocampal neurons, that can be partially reversed by oral Zn supplementation of the pups after weaning [113]. Studying these molecular abnormal events occurring during development is complicated in humans, thus the reliance on animal models. However, because the brain also develops postnatally and has an indispensable plasticity during childhood and adolescence, some results obtained during these periods in humans can be very informative concerning the early consequences of Zn deficiency. Indeed, the impact of Zn deprivation during childhood was studied by Golub and coworkers, who showed that moderate Zn deprivation can cause a decrease in short-term memory and attention, as well as a reduction in motor activity [114,115,116,117], consistent with results obtained with animal models described above [111,112,118]. The molecular underpinnings of how early Zn deficiency produces long-term neurological defects still remain poorly understood, although some progress has been made in this area, which we describe below.

The molecular consequences of developmental Zn deficiency on the hippocampus appear to involve dysregulation of the TrkB receptor, independent of its ligand BDNF (Brain-Derived Neurotrophic Factor). Indeed, pups from dams subjected to Zn deficiency suffer from decreased levels of TrkB phosphorylation, leading to reduced Erk1/2 signaling pathway and the activation of caspase-dependent apoptosis through Bax/Bcl2 [119]. In the cerebral cortex and ventricular zone of fetal brain of rats fed a marginally deficient Zn diet, the downregulation of the Erk1/2 signaling pathway was also observed, which was attributed to increased activity of PP2A phosphatase [120]. Interestingly, these molecular disruptions of the Erk1/2 signaling pathway are associated with a lower density of neurons in fetal brain, and these alterations persist into adulthood in the same rat model, suggesting that some of the long-term consequences of early Zn deficiency may be connected to alterations in Erk1/2 signaling [121]. However, other signaling pathways are also affected. Experiments based on human neuroblasts cultured in vitro demonstrated impaired nuclear translocation of NF-kB under Zn deprivation conditions [122]. These results may explain why apoptosis occurs in fetal brain in response to Zn deficiency during brain development, as described above. Other transcription factors have also been reported to be dysregulated in response to Zn deficiency conditions during development; AP-1 and N-FAT display alterations of nuclear translocation, DNA binding, and pathway activation in the fetal brain of rats fed a gestational, Zn-deficient diet [123]. STAT1 and STAT3 transcription factors are known to play major roles in CNS development through their role in cell proliferation, cell death, survival, and differentiation [124,125,126,127]. Using Zn-depleted IMR-32 neuroblastoma cells, the authors also showed a decrease in STAT1- and STAT3-dependent gene transactivation, as well as an increase in oxidative stress markers and abnormalities in cytoskeleton dynamics in both IMR-32 cells and fetal brains, demonstrating that changes in the cytoskeleton are also observed in response to Zn deficiency [128]. In addition, a disruption of STAT1 and STAT3 signaling pathways activation can also be observed in response to marginal maternal Zn deficiency in fetal rat brain [128]. The disruption of STAT1 and STAT3 signaling was associated with an increase in oxidative stress markers, as well as abnormalities in cytoskeleton dynamics, demonstrating that changes in the cytoskeleton are also observed in response to Zn deficiency. Indeed, decreased expression of Nestin, a type VI intermediate filament cytoskeletal protein, was observed in the cerebral cortex and the neural tube during early stages of development in a maternal Zn deficiency mouse model [129]. Changes to other cytoskeletal proteins have also been observed. Oteiza and colleagues detected a decreased rate of tubulin polymerization in the fetal brain of pups challenged with severe maternal Zn deficiency before birth, which could be reverted in vitro by Zn supplementation of the culture medium of brain supernates [130]. Myelination is also altered by early maternal Zn deficiency. Liu and colleagues showed modifications of the protein composition of myelin in the brain of rats and monkeys submitted to a mild Zn deficiency [131]. Zn deficiency also leads to changes in the expression of NMDA receptor subunits, with a decrease in NR1, NR2A, and NR2B in the whole brain and hippocampus at P2 and P11 in pups from dams fed a Zn-deficient diet [118]. Moreover, whole brain NR1 expression remained lower in previously zinc-deficient rats at PN65 [118]. These changes are accompanied with a decrease in brain NGF (Nerve Growth Factor), that persists for two months after birth.

Lastly, the molecular mechanisms responsible for the pathogenesis of ASD in an early Zn deficiency context have been partly elucidated by Grabucker and colleagues. Using an acute prenatal maternal Zn deficiency mouse model, they showed that postsynaptic density scaffolding at the synapse was regulated by Zn levels, and that Zn deficiency leads to the dysregulation of synaptic levels of the scaffolding proteins ProSAP/Shank in vivo [132]. Furthermore, these molecular alterations were linked with neurobehavioral abnormalities, such as impairments in vocalization and social behavior. It is worth noting that the patients suffering from Phelan McDermid Syndrome and exhibiting autism-related symptoms caused by a deletion of Shank3 scaffolding protein (22q13.3 deletion) also have lower expressions of the ZIP2 Zn transporter in enterocytes [133]. Another study from the same research team also showed that Shank3 transgenic mouse lines and prenatal Zn deficiency ASD models have similar phenotypes, especially concerning alterations in brain structures, including an increased size of the basal ganglia structures [134]. Thus, changes in Zn status in early life can have complex effects on brain development and function whose origin stems from the dysregulation of many different proteins and signaling pathways. Although much progress has been made in understanding the impact of gestational Zn deficiency on the structure and function of the brain, much research remains to be done.

## 5. Impact of Zinc Transporter and Channels on Zinc Deficiency and Development

While it is clear that Zn plays a critical role in systemic homeostasis and development, how Zn levels are maintained throughout different organ systems and the cells within them remains poorly understood. Nor is it clear what the function of Zn is during development and how the critical cation homeostasis is maintained. A total of 24 genes encoding Zn transporters have been identified so far. The corresponding proteins are divided in two groups: the ZnTs (Table 3), composed of 10 members and encoded by *SLC30A-(1-10)* genes; and the ZIPs (Table 4), composed of 14 members and encoded by SLC39A-(1-14) genes [135]. ZnTs and ZIPs proteins display distinct and specific patterns of expression within the cell, allowing the transport of Zn ions between the different cellular compartments and therefore participating in cellular Zn homeostasis. While ZIP proteins can mediate the transport of Zn, they can also transport other ions into the cells, including manganese (Mn), cobalt (Co), and cadmium (Cd) into the cell [136]. Therefore, mutations to members of either gene family of Zn transporters can produce deleterious phenotypes in human and animals, with pathological effects comparable to some symptoms of dietary Zn deficiency. Below we review what is known regarding the expression, cellular localization, and function of these transporters in adult animals and during development, and introduce the TRPM7 channel, a new protein recently found to contribute to systemic Zn homeostasis.

### 5.1. ZnTs Transporters: SLC30A Family

#### 5.1.1. ZnT1 (SLC30A1)

In 1995, ZnT1 (Table 3) was discovered as the first member of the ZnT family and was found to function as a Zn exporter [137]. While ZnT1 protein is expressed ubiquitously [138], some tissue-specific features have been uncovered, such as a protective role against Zn toxicity in the brain [239]. In the polarized epithelial cells lining the intestine, ZnT1 is found in enterocytes at the basolateral surface, where it is responsible for Zn efflux to the global circulation [137]. Interestingly, the *Drosophila* ortholog of ZnT1 (dZnT1) has been demonstrated to regulate Zn absorption, but similar results have not been reported in mammals [240]. Knockout of ZnT1 in mice leads early embryonic lethality with in utero death after implantation before embryonic day nine, with embryos exhibiting defects in morphogenesis as soon as embryonic day seven [139]. The same study also showed that heterozygous KO of ZnT1 in mouse embryos renders them more susceptible to developmental defects in response to maternal Zn deficiency during later pregnancy compared to those in wild-type females, suggesting additional undeciphered roles of ZnT1 during embryogenesis and more generally in development [139].

#### 5.1.2. ZnT2 (SLC30A2)

The ZnT2 protein (Table 3) is strongly expressed in endocrine tissues, such as the pancreas and the mammary gland [138,142]. The transporter is localized to vacuoles and lysosomes [140], with a putative role in Zn sequestration [241]. *SLC30A2* transcript levels in rat small intestine are confined to enterocytes and become highly expressed postnatally [145]. However, similar results have not been reported in human. In mouse mammary gland, ZnT2 protein expression is reported to increase during Zn deficiency [242], whereas in rat small intestine and kidney, *SLC30A2* transcript expression is increased during Zn supplementation, suggesting a complex tissue-dependent mechanism of regulation of protein expression [160].

Interestingly, mutations in *SLC30A2* in humans lead to offspring TIZD, a pathology causing Zn deficiency in newborns under six months that are exclusively breastfed [147]. Several mutations leading to TIZD have since been discovered, including the G87R dominant negative heterozygous [243] mutation and the homozygous 262G > A (p.E88K) mutation. The latter mutation results in a more significant decrease in Zn secretion and a more rapid onset of TIZD [244] and is the most common mutation found in humans [245]. These results have been reinforced by a milk-derived miRNAs (microRNAs) profiling study, which showed that different *SLC30A2* mutations lead to various lactation performances. Moreover, this profiling study also underscored alterations of numerous molecular pathways, with almost 7000 predicted mRNA targets, and 137 Gene Ontology (GO) categories, such as “organelle function,” “ion binding,” “RNA binding,” “biosynthetic processes,” “cellular component assembly,” “catabolic process,” and “response to stress” (authors did not communicate the unique GO IDs corresponding to these categories) [246]. The genetic variant T288S has also been widely studied and some recent results suggest that this specific mutation causes sequestration of the ZnT2 protein to the ER and lysosomes, where the mutated protein is reported to cause activation of STAT3 pathway through an increase in oxidative and ER stress [247,248,249].

#### 5.1.3. ZnT3 (SLC30A3)

In mouse and human, ZnT3 (Table 3) transcript has been shown to be enriched in brain and testis [150], and the human ZnT3 protein has been shown to have high levels of expression in testis and epididymis [138]. In the brain, ZnT3 is mainly expressed in glutamatergic neurons of the hippocampus [152] and the cerebral cortex, where the transporter mediates Zn uptake into synaptic vesicles [151]. ZnT3 is the only neuron-specific isoform present on synaptic vesicles, as well as the only known route by which Zn ions are loaded into synaptic vesicles of a subset of glutamate neurons in the brain [149]. Thus, KO experiments have demonstrated the importance of ZnT3 for synaptic vesicular Zn transport in mouse, especially in the cortex and hippocampus [148], which has been extensively reviewed by McAllister and Dyck [149]. However, ZnT3 KO mice do not show any obvious developmental abnormality and are broadly normal, both behaviorally and electrophysiologically [149]. In addition, ZnT3 also has a role in the maintenance of neurogenesis in adult hippocampus, where it regulates cell proliferation and neuronal differentiation in the subgranular zone of the dentate gyrus [250]. ZnT3 KO mice also exhibit behavioral defects, especially concerning spatial memory [154], which could be explained by a reduced activation of the Erk1/2 MAPK (Mitogen-Activated Protein Kinase) in hippocampal mossy fibers [155]. Interestingly, these characteristics seem to be regulated in a sex-dependent manner [251], with female ZnT3 KO mice exhibiting increased locomotion defects but lower cognition and learning deficits compared to males [153,154]. While no diseases causing mutations in humans have been reported, a meta-analysis found that Single Nucleotide Polymorphisms (SNPs) mutations in *SLC30A3* gene are associated with schizophrenia in female but not male individuals in European populations [252,253].

Studies have also shown that protein expression of ZnT3 is altered during aging. Using an accelerated aging mouse model, Saito and coworkers showed a reduction in ZnT3 expression in hippocampal neurons that was associated with abnormal presynaptic glutamate release and cell injury markers [254]. Work by Hancock and colleagues showed that KO of ZnT3 leads to alterations of the metalloproteome, which consists of specific metal binding proteins, in an age-dependent manner [255]. Lastly, ZnT3 has been widely associated with deleterious phenotypes, such as seizures [256], brain injuries [257,258], as well as neurodegenerative diseases, including Amyotrophic Lateral Sclerosis (ALS) [259], Huntington’s disease (HD) [260], Parkinson’s disease (PD) [261], and AD [262,263,264,265].

#### 5.1.4. ZnT4 (SLC30A4)

ZnT4 (Table 3) is widely expressed in humans, with notable enrichment in the brain (hippocampus and caudate), as well as several other organs, including the thyroid, lung, testis, heart, skin and the pancreas [138]. Expression of ZnT4 in the mammary gland has also been reported in mouse, where it has been shown to cause the milk lethal phenotype, causing similar symptoms to TIZD [158,159]. It is nevertheless important to note TIZD is not caused by *SLC30A4* mutations, but instead originates from *SLC30A2* loss of function [266]. Thus, in mouse, ZnT4-deficient dams produce milk with low Zn content, leading to postnatal Zn deficiency in the offspring. Additionally, this phenotype can be reverted when pups are supplemented with Zn or fostered by wild-type dams [159,267,268]. Interestingly, Zn dietary restriction has been shown to cause an increase in ZIP4 expression in small intestine [196]. Disabling mutations in the ZnT4 Zn transporter cause a decrease in Zn secreted in breast milk [161], which will lead to an early Zn deficiency in pups fed by a dam carrying this mutation and eventually to their premature death before weaning [159]. The Zn deficiency symptoms observed in pups harboring ZnT4 mutations can be reverted by Zn supplementation, or by putting these pups with wild-type dams [159,267]. Interestingly, even after Zn supplementation, ZnT4 mutant pups fed by wild-type dams will develop balance and spatial memory behavioral abnormalities [267].

Lastly, although *SLC30A4* transcripts have also been detected in human [138] and mouse small intestine [160], no evidence of the presence of ZnT4 protein in this organ has been reported. Moreover, young ZnT4 KO mice show normal Zn absorption, suggesting that other Zn transporters compensate for the loss of ZnT4 in mice.

#### 5.1.5. ZnT5 (SLC30A5)

ZnT5 (Table 3) is ubiquitously expressed [138], with high levels of expression in parietal cells of the stomach and in the absorptive epithelium of the duodenum and jejunum [166]. The ZnT5 protein is localized to the Golgi apparatus [162] and can be found in cytoplasmic vesicles [269]. It has been previously reported that a high Zn dietary intake leads to a decrease in both *SLC30A5* transcript and protein levels in human small intestine [270]. Interestingly, a team studied the different splice variants of *SLC30A5* transcripts and showed that they encode for proteins with different subcellular localizations [163,164]. Using *Xenopus* oocytes and the Caco-2 intestinal cell line, the researchers showed that the human ZnT5 variant B (ZnT5B (hZTL1)) is localized to the plasma membrane, where it functions as a Zn exporter, similarly to other *SLC30A* family members, but also as a Zn uptake regulator [271]. To the best of our knowledge, no other member of the ZnT or ZIP families possesses this dual function in mammals. However, bidirectional transporter-mediated transport of Zn is not without precedent and was reported in *Saccharomyces cerevisiae* for the transporter Yke4p, a member of the SLC39 family [272]. Additionally, the transcriptional expression of both ZnT5A and ZnT5B splice variants is decreased under Zn deficiency, as well as in response to Zn supplementation, suggesting a role for the ZnT5 bidirectional transporter in fine regulation of Zn uptake at the cellular level [163]. However, different experimental results were obtained by another team, who showed that Zn supplementation of HeLa cells does not lead to any changes in *SLC30A5* mRNA levels, whereas Zn depletion with TPEN causes an increase in transcriptional expression of ZnT5 expression [273]. Nevertheless, it is important to note that these two studies do not use the same experimental approach in terms of Zn supplementation and depletion, nor did they use the same cell line or the same gene expression measurement method, pointing to the need for additional study. Interestingly, pathological variants of *SLC30A5* that cause homozygous loss of function have been shown to cause cardiomyopathy and perinatal death [167], similar with what has been reported for ZnT5 KO mice [168]. However, while ZnT5-deficient mice show growth defects as well as osteopenia, muscle weakness, and male-specific cardiac death, they do not suffer from global Zn deficiency, showing that ZnT5 is not responsible for Zn absorption [168]. Lastly, ZnT5 has also been associated with various pathological phenotypes in humans, including an increase in inflammatory markers [274], obesity [275,276], type 2 diabetes mellitus (T2D) [277], colorectal cancer [278] and AD [279].

#### 5.1.6. ZnT6 (SLC30A6)

In humans, ZnT6 (Table 3) is ubiquitously expressed [138]. In mouse, ZnT6 is also widely expressed, with predominant expression in the brain, lung, and the small intestine [169], and more specifically in the duodenum and jejunum [166]. A comparative study from Overbeck et al. showed that ZnT6 transcriptional expression was increased under Zn deficiency, induced by TPEN treatment in different leukocyte subsets, including in Raji (B-cell line) and THP-1 (monocyte) cells, but not in Molt-4 (T cell line), or in freshly isolated peripheral blood mono-nuclear cells (PBMCs) [280]. Interestingly, *SLC30A6* KO experiments have been performed in the chicken lymphoblast DT40 cell line [165], which demonstrated the importance of ZnT6 for the activation of alkaline phosphatase. ZnT6 has been linked to various detrimental phenotypes and pathologies in humans, including mild cognitive impairment [281,282], ALS [259], Fragile X-associated tremor/ataxia syndrome (FXTAS) [283,284], early [282,285,286] and late [262,282] models of AD, and pancreatic cancer [287]. However, as of this writing, no ZnT6 KO animal model has been reported. Thus, the importance of ZnT6 during development is not known.

#### 5.1.7. ZnT7 (SLC30A7)

ZnT7 (Table 3) is ubiquitously expressed in humans, exhibiting significantly high protein expression in the small intestine and colon and comparatively low expression in the brain [138]. In mice, high *SLC30A7* transcript expression has been reported in liver, kidney, spleen, and the small intestine [170]; however, ZnT7 protein expression was only detected in the lung and the small intestine. Additionally, another study showed ZnT7 was observed in epithelial cells of the mouse gastrointestinal tract with the highest expression in the small intestine [166]. Interestingly, KO of *SCL30A7* in mice leads to smaller weight and retarded growth of the offspring that cannot be reverted by dietary Zn supplementation [171]. This deleterious phenotype is associated with a global decrease in serum Zn levels as well as low liver, duodenum, kidney, and bone Zn content. Surprisingly, these mice do not exhibit typical symptoms of Zn deficiency, such as alopecia or dermatitis, indicating the need for more study to understand how ZnT7 contributes to Zn deficiency pathophysiology. Lastly, ZnT7 may be involved in cancer. High ZnT7 expression is associated with colorectal cancer in humans [278], while low ZnT7 levels seem to be related to tumor growth in *Drosophila* [288] and prostate cancer in mice [289].

#### 5.1.8. ZnT8 (SLC30A8)

ZnT8 (Table 3) is almost solely expressed in the pancreas in humans [172]. However, little is known concerning the control of ZnT8 expression. Interestingly, ZnT8 can increase insulin production and excretion [175,176,290,291,292]. Thus, in human pancreatic islet cells, ZNT8 is exclusively expressed in insulin-producing beta cells, and colocalized with insulin in these cells. When overexpressed, ZnT8 stimulates Zn accumulation in the pancreas and leads to enhanced glucose-stimulated insulin secretion [290]. Conversely, deletion of ZnT8 from mice leads to global pancreatic function defects as well as low Zn content in pancreatic tissue [172,173,174,175,176]. Consequently, ZnT8 function has been extensively studied in both type 1 [293,294] and type II [295,296] diabetes (T1D and T2D, respectively) in various populations [297,298,299,300,301]. Interestingly, a study showed that the levels of methylation of *SLC30A8* promoter were correlated with T2D in a sub-Malaysian population [302]. Promoter methylation is generally associated with gene silencing, and this potential regulation of SLC30A8 expression and its connection to diabetes should continue to be studied. In addition, multiple *SLC30A8* polymorphisms and SNPs have been identified over the two last decades, including the most widespread rs13266634 C/T [303,304,305], also referred to as Arg325Trp [306,307], rs11558471 [308,309], and rs11203203 [310]. Carvalho and colleagues demonstrated that distinct splice and SNPs variants in *SLC30A8* can result in different subcellular localization of the expressed ZnT8 protein when studied in vitro, with some variants of the transporter localized at the cell surface and others to intracellular membranes [311]. This result may be due to the fact that the different splice and SNP variants alter the structure of the ZnT8 protein. Elucidating the functional impact of the different ZnT8 protein variants is essential for understanding how they affect diabetes [312,313,314,315]. New specific antibodies are also required for immunodetection of these variants [316,317,318,319,320]. In addition, more functional studies of the variants are needed to develop specific strategies to counteract their contribution to diabetes pathology [321].

Intriguingly, the loss of function of ZnT8 in humans and mice is associated with protective effects against T2D [322,323,324] and T1D [325] in vitro and in vivo. However, even if an inhibition of ZnT8 could be a promising therapeutical strategy in diabetes, these results remain controversial since other studies showed that inhibition of ZnT8 has no effect on glucose-stimulated insulin secretion [326]. Moreover, ZnT8 depletion targeting has been shown to produce detrimental consequences [327], including, but not limited to, increased sensitivity to a high-fat diet [174], obesity susceptibility [328], impaired glucose tolerance, and increased basal insulin clearance [329], and elevated circulating serotonin (5-hydroxytryptamine) [177]. These adverse consequences of loss of ZnT8 are reviewed in [330], and notably occur through potential extra-pancreatic function of ZnT8 that does not seem to be compensated by ZnT7 [331]. Altogether, these results show potential clinical benefits of targeting ZnT8 in diabetes, but also negative effects associated with T2D prevention using ZnT8 inhibitors.

In addition to its effects in pancreas, it was reported that KO of ZnT8 in mice produced increased adiposity, manifestly through the control of serotonin (5-hydroxytryptamine) synthesis, underlining then a potential role of ZnT8 in control of obesity [177] and metabolic syndrome [332] and the importance of developing therapeutics targeting ZnT8 not only in diabetes. Thus, more research is needed to understand how expression of *SCL30A8* is regulated and how different mutations affect ZnT8 protein structure and function, which will be critical to make progress toward developing novel treatment strategies targeting ZnT8 [333].

#### 5.1.9. ZnT9 (SLC30A9)

ZnT9 (Table 3) was first named HUEL (Human Embryonic Lung Protein) and is expressed ubiquitously in humans [138], with highest levels detected in cerebellum, skeletal muscle, thymus, and kidney [181]. Interestingly, regulation of ZntT9 expression appears to be dependent on sex hormones, since both *SLC30A9* transcript and protein expression have been shown to be drastically decreased in vaginal tissues after menopause [334]. In cells, ZnT9 has been detected in the cytoplasm, but in some cell lines, such as PLC/PRF/5 and TONG liver carcinoma cells, the transporter can be found in the nucleus [179,180] where it has been proposed to have a role as a nuclear co-activator [178]. Thus, HUEL-ZnT9 was also named GAC63 for GRIP1-associated coactivator 63. However, more recent studies, employing computational and loss of function analyses, have found a preponderant role of ZnT9 in the regulation of mitochondrial Zn homeostasis, which is conserved throughout evolution [335,336].

Despite this progress, the function of ZnT9 and its regulation remain poorly understood. Knockout of ZntT9 in mice has not yet been pursued, and thus, the transporter’s contribution to development is not known. However, a mutation in the *SLC30A9* gene, consisting of the deletion of three nucleotides (c.1047_1049delGCA, p.(A350del)), was found in six individuals of consanguineous Bedouin kindred [181]. This specific mutation is predicted to lead to structural changes in the ZnT9 protein that disrupt transport activity, consistent with the reduction in the intracellular levels of Zn that were measured in vitro. The affected individuals showed neurological alterations progressing into severe intellectual disability, ataxia, and profound alterations of renal function, suggestive of a novel autosomal recessive cerebro-renal syndrome. Finally, ZnT9 expression also has been associated with other deleterious phenotypes and pathologies, including hepatocellular carcinoma [337] and prostate cancer [338].

#### 5.1.10. ZnT10 (SLC30A10)

ZnT10 (Table 3) is expressed in the small intestine as well as in the liver and cerebral cortex [138]. Furthermore, its transcript has been found overexpressed in liver cancer, and ZnT10 protein levels are increased in colorectal, liver, and endometrial cancers [339]. Using an in vitro cell model, Bosomworth and colleagues showed that supplementation of Zn in cell culture media produced a change in ZnT10 localization from the Golgi to the plasma membrane [182]. Expression of *SLC30A10* also seems to be negatively regulated by Zn, since intestinal and neuroblastoma cell lines challenged with high Zn concentration in cell culture media lower transcript levels. Taken together, these results suggest a Zn transporter role for ZnT10.

Nevertheless, ZnT10 has been shown to be involved in Mn transport. Indeed, the second transmembrane domain of ZnT10 contains a non-conserved asparagine residue instead of a histidine residue that is found in other ZnTs, which could explain ZnT10 selectivity for Mn over Zn [340]. This would also explain why mutations in *SLC30A10* gene give rise to high blood Mn levels in humans [184,185] and Mn-induced neurotoxicity [341], and are associated with AD [342] and parkinsonism [343,344]. Consistent with the human phenotype, ZnT10 KO mice exhibit no changes in blood plasma or tissue Zn levels, but do suffer from high Mn content in plasma, brain, and liver [183]. ZnT10 KO mice were smaller in size than littermate controls and developed hypothyroidism, but do not otherwise exhibit any obvious developmental abnormality [183,345,346].

### 5.2. ZnTs Transporters: SLC30A Family

#### 5.2.1. ZIP1 (SLC39A1)

*SLC39A1* (Table 4) transcript expression is ubiquitous in humans [138,186], with notably high levels in small intestine [192] and in the brain, including the hippocampus, thalamus, and the cerebral cortex [189]. ZIP1 has been shown to localize to the plasma membrane, where it mediates energy-independent cellular Zn uptake [186]. However, overexpression of ZIP1 in several cell lines has revealed an ER [192], and, alternatively, vesicular localization [188]. Site-directed mutagenesis studies have further shown that ZIP1 can be localized to the plasma membrane and is internalized by endocytosis through a di-leucine motif (ETRALL144–149) [187].

While ZIP1 transcript expression is increased ubiquitously during late fetal development in mouse [347], ZIP1 KO mice do not exhibit any obvious developmental defect [190] However, when ZIP KO mice are challenged a Zn-deficient diet, embryos exhibit abnormalities, ranging from mild to severe. A high proportion of embryos had a mild retarded growth, with delayed development of the hindlimb digits but visible digits on the forelimbs. However, more severe defects were observed in other animals, showing severe growth retardation, no obvious digits forming on either the forelimbs or hindlimbs, and craniofacial defects, which were ultimately embryonically lethal [190]. These results suggest an important role for ZIP1 in Zn deficiency adaptation rather than a direct involvement in Zn absorption [191,192]. Consistent with this hypothesis, Zn supplementation led to a diminution of *SCL39A1* transcriptional expression in primary cultures of adult rat ventricular myocytes [348]. Comparable results have been obtained in humans, with some female adolescent showing an increase in *SLC39A1* mRNA expression in blood leucocytes following Zn supplementation with Zn-fortified milk for 27 days [349]. While an increase in *SLC39A1* mRNA levels were not observed in all individuals from the treated group [349], comparable results have been obtained in humans by another group, demonstrating an increased ZIP1 mRNA expression in lymphocytes in an Australian elderly population supplemented with Zn [350]. Lastly, ZIP1 is associated with various cancers [351,352,353], including pancreatic [287] and prostate [354,355,356,357] cancer.

#### 5.2.2. ZIP2 (SLC39A2)

ZIP2 (Table 4) is broadly expressed in humans, found in endocrine tissues, lungs, gastrointestinal tract, liver, pancreas, kidneys, heart, and different brain structures, including the cortex, hippocampus, cerebellum, and the basal ganglia [138]. Expression of mouse ZIP2 (mZIP2) in the HEK-293 cell line (Human Embryonic Kidney 293) demonstrated the transporter’s Zn uptake activity [199]. Consistent with its role in cellular Zn homeostasis, both ZIP2 mRNA and protein expression levels are upregulated in monocyte cell line THP-1 under Zn depleted conditions induced by the Zn-chelator TPEN [358]. Conversely, the same team observed a downregulation of ZIP2 mRNA expression by microarray analysis in the same THP-1 cell line supplemented with Zn [359]. Surprisingly, Zn dietary deficiency in mice does not lead to variation in intestine ZIP2 expression, suggesting that the protein has a more specific a role in cellular Zn homeostasis, rather than in intestinal Zn absorption [199].

Interestingly, deletion of *SLC39A2* in mice does not produce to any obvious developmental defects [193]. However, authors also showed that when ZIP2 KO mice are fed a Zn-deficient diet, important developmental abnormalities are observed in offspring. Indeed, the authors reported 57% of embryos from ZIP2 KO dams fed a Zn-deficient diet had abnormal limb differentiation and cranio-facial abnormalities, compared to 14% of wild-type animals, suggesting a role for ZIP2 in Zn deficiency adaptation during development.

Beyond its role in development, high expression of ZIP2 is associated with inflammatory conditions, including asthma [360] and pulmonary infection [361] in humans. Similarly, the ZIP2 Gln/Arg/Leu (rs2234632) polymorphism is associated with severe carotid artery disease [362], especially under low Zn conditions [363]. Lastly, it is also worth noting that deletion of the Shank3 scaffolding protein leads to a decrease in ZIP2 expression in enterocytes in patients exhibiting autism-related symptoms [133]. This observation is consistent with previous results from the same investigatory team in which they showed that prenatal and acute Zn deficiency caused dysregulation of the scaffolding proteins ProSAP/Shank at the synaptic level in mice [132].

#### 5.2.3. ZIP3 (SLC39A3)

In humans, ZIP3 (Table 4) protein expression is high and almost ubiquitous, with particularly elevated levels along the gastrointestinal tract [138]. When overexpressed in HEK-293 cells, murine ZIP3 has peri-nuclear localization in the *trans*-Golgi apparatus [194]. Interestingly, the localization of ZIP3 is altered under Zn depletion conditions, which stimulates recruitment of ZIP3 to the plasma membrane [194].

Regulation of ZIP3 expression in response to changes in Zn availability remains unclear. Cousins and co-workers showed that Zn depletion with TPEN in the THP-1 monocyte cell line led to downregulation of *SLC39A3* mRNA levels [359]. However, a study by Dufner-Beattie and colleagues found no variation in intestinal ZIP3 mRNA expression in mice subjected to dietary Zn deficiency, similar to what was found for ZIP2 [199].

ZIP3 is not essential for development since ablation of the gene in mice does not lead to any obvious developmental abnormalities [195]. However, a higher proportion of embryos exhibit growth retardation or limb bud and craniofacial defects when subjected to maternal Zn deficiency. Pups from ZIP3 KO dams fed a Zn-deficient diet also exhibit decreased thymic and testicular weight at weaning. Surprisingly, while ZIP1 and ZIP3 double KO in mice demonstrated no deleterious developmental phenotype under normal conditions [190], challenging the double KO mice with a Zn-deficient diet causes embryos to present with higher rates of mild (growth retardation with delayed development of the hindlimb digits, but visible digits on the forelimbs) and severe (growth retardation, no obvious digits forming on either the forelimbs or hindlimbs, craniofacial defects) developmental abnormalities compared to non-treated ZIP1 KO mice. These results demonstrate an important role for both ZIP1 and ZIP3 in Zn deficiency adaptation during development.

The apparent redundancy in ZIP transporter functions as well as their role in Zn deficiency adaptation have been further investigated and highlighted by Kambe and coworkers. Strikingly, ZIP1, ZIP2, and ZIP3 triple KO mice had no visible adverse consequence on pregnancy rate, litter size, or embryo growth and development under normal conditions, but led to approximately 80% of pups having developmental abnormalities at day 14 under Zn deficiency conditions, with about 60% having severe defects (very small embryos with abnormal forelimbs and hindlimbs, as well as craniofacial abnormalities) [191].

#### 5.2.4. ZIP4 (SLC39A4)

ZIP4 (Table 4) is considered to be the primary mammalian Zn uptake transporter in the small intestine. Thus, mutations in *SLC39A4* give rise to *Acrodermatitis enteropathica* (*AE*), a rare autosomal, recessively inherited disorder of intestinal Zn malabsorption that is also called “genetic Zn deficiency” [23,54,55,56]. Physiologically, levels of the *SLC39A4* transcript show a low, but nevertheless detectable, tissue-specific expression in the kidney and colon, with higher levels found the in duodenum and jejunum, the mains sites of Zn absorption [23,56,138]. *SLC39A4* transcript expression also has been marginally detected in other tissues and cell types, such as mouse microglial BV-2 cell line [364] and different rat brain structures, including choroid plexus and ventricles [189]. Both ZIP4 mRNA and protein have also been detected in vitro in rat astrocytes and neurons at postsynaptic sites [201].

In enterocytes, ZIP4 is localized to intracellular vesicles, but can be recruited to the plasma membrane under Zn deficiency conditions, which is accompanied by an increase in Zn uptake in vitro and in vivo [196,197,198,200]. ZIP4 accumulation in the small intestine of rat challenged with Zn-deficient diet has been shown to occur as early as one day following the Zn-deficient diet, indicating that ZIP4 could potentially be used as an early marker of Zn deficiency in neonates [365]. Indeed, *SLC39A4* transcript levels are decreased in the blood of children suffering from Zn deficiency, prompting authors of a recent study to propose using ZIP4 as a marker of Zn deficiency [366]. It is also worth noting that Zn supplementation leads to the decrease in ZIP4 mRNA expression in human ileum [270]. However, the initial Zn statuses of these subjects were not been measured and the study does not presume a potential reversion of Zn-deficient phenotype by Zn supplementation. Nevertheless, regulation of ZIP4 expression in the human intestine in response to variations in dietary Zn intakes contributes to maintenance of Zn status [270]. Indeed, for almost two decades it has been known that mutations in the *SLC39A4* gene cause *AE* [23,54,55,56]. *AE* is the most known and studied genetic form of Zn deficiency. It is an autosomal recessive disease affecting 1:500,000 newborns, causing a defect in Zn absorption [57]. Although *AE* was initially described by Brandt in 1936, it acquired its name by Danbolt in 1948 [367]. However, *SLC39A4* the mutated gene responsible for *AE* was first identified in 2002 [56]. Patients suffering from *AE* present with defects in Zn absorption and thus manifest clinical signs of Zn deficiency, including as periorificial and acral dermatitis, alopecia, and diarrhea. While *AE* is transmitted in an autosomal recessive mode in humans, studies have shown that knockout of ZIP4 in mice is embryonically lethal during the post-implantation period [202]. Heterozygous ZIP-4 KO mice display hypersensitivity to maternal Zn deficiency. MRI (Magnetic Resonance Imaging) imaging also showed that ZIP4^+/−^ mice show global growth retardation, as well as brain and heart development defects. Indeed, heterozygous KO mice display high rates of developmental abnormalities, including exencephaly and severe growth retardation. Furthermore, the same study also demonstrated that heterozygous embryos from wild-type dams are more sensitive to maternal Zn deficiency. Altogether, these results demonstrate that both maternal and embryonic ZIP4 are essential for Zn homeostasis during development. Lastly, conditional ZIP4 KO was used to show that deletion of ZIP4 in intestinal epithelium led to the disruption of the intestinal stem cell niche, followed by the disorganization of the absorptive epithelium and the disruption of its function [203]. Beyond its impact on Zn homeostasis, studies of ZIP4 have also revealed a role for the transporter in the transport of copper and nickel [368], organization of the intestinal epithelium [203], and also in pathophysiological contexts, such as cancer [369,370]. Despite the discovery of the gene responsible for *AE* in human almost 20 years ago, the molecular function and regulation of ZIP4, as well as the consequences of its dysfunction, still remain poorly understood [56] and further studies will be a key to deciphering its broad functions during development and in adults.

#### 5.2.5. ZIP5 (SLC39A5)

ZIP5 (Table 4) has a similar expression pattern to ZIP4. *SLC39A5* mRNA is expressed along gastrointestinal tract as well as in the kidney, the two major sites of Zn absorption in the body [138,204]. ZIP5 has also been shown to be expressed in the liver and in the pancreas [204]. Another study found that ZIP5 could also be detected in human peripheral blood mononuclear cells [206]. When HA-tagged version of murine ZIP5 (mZIP5) was overexpressed in MDCK canine polarized kidney cells, the transporter localized to the basolateral pole of the cells [204]. The native protein showed a similar cell distribution in vivo when studied in mouse pancreatic acinar cells and intestinal enterocytes and endoderm cells [200]. Similar to ZIP4, ZIP5 has been shown to be involved in the adaptative response to Zn deficiency [200]. It was found that ZIP5 is internalized from the basolateral membrane in enterocytes, acinar cells, and endoderm cells in response to Zn deficiency [200,205]. Interestingly, ZIP5 deletion in mouse is not lethal and does not cause developmental abnormalities; instead, loss of the ZIP5 in mice leads to Zn accumulation in the pancreas and in the liver, causing Zn-induced pancreatitis [207]. Altogether, these results suggest that ZIP5 is involved in: (1) Zn transport from the blood stream into enterocytes; (2) Zn elimination to avoid its toxicity, and; (3) Adaptation to Zn deficiency, by being internalized and consequently increasing the intestinal uptake of Zn, while ZIP4 expression is increased in the meantime. 

#### 5.2.6. ZIP6 (SLC39A6)

ZIP6 (Table 4) was first identified as an “estrogen induced gene” called LIV-1 at the end of the 1980s in ZR-75-1 [371] and MCF-7 [372,373,374] breast cancer lines, as well as in vivo in human breast tumors [375]. It was only at the beginning of the 2000s that computational studies highlighted ZIP6′s potential Zn transporter function [208,376]. ZIP6 mRNA and protein expression are ubiquitous in humans with notable high levels in the cerebellum, adrenal gland, and endometrium [138]. Overexpression of a V5-tagged version of ZIP6 in CHO cells (Chinese-Hamster Ovary cell line) demonstrated that the protein localized to the plasma membrane [208]. ZIP6 can also form a heterodimer with ZIP10; such oligomerization is an uncommon feature in the ZIP protein family [377,378]. Both ZIP6 and ZIP10 display a strong reduction in expression in testis of male mice fed a severe Zn-deficient diet compared to males fed a marginal Zn-deficient diet, or a normal diet [379]. Moreover, recent results showed the importance of the heterodimer ZIP6/ZIP10 during mitosis, when it mediates the import of Zn into the cells [377].

ZIP6 has important functions in the central nervous system (CNS), where it is expressed in most of the brain structures. Chowanadisai and coworkers demonstrated that maternal Zn deficiency in rat increased expression of both ZIP6 transcript and protein in the offspring’s brain, and more specifically in the hippocampus [380]. Conversely, the same team showed that Zn supplementation of SH-SY5Y neuroblastoma cells decreased ZIP6 protein expression [381]. However, the authors did not report any developmental abnormalities in ZIP6 KO rats. Using a siRNA approach, ZIP6 was shown to be required for Zn uptake in vitro in SH-SY5Y neuroblastoma cells. Interestingly, these results have been obtained under resting conditions, but not under depolarizing conditions, suggesting the existence of at least two different, non-redundant neuronal Zn-uptake systems functioning complementarily.

Besides its role in cancer progression, ZIP6 may play a role in AD pathogenesis. Postmortem brain biopsies of AD patients revealed increased expression of the *SLC39A6* transcript in the superior frontal gyrus [262], as well as increased ZIP6 protein levels in the hippocampus and parahippocampal gyrus [281]. Lastly, ZIP6 has also been demonstrated to be involved in the immune response, in which it is essential for T-cell activation in humans [382,383] and in Zebrafish [384].

#### 5.2.7. ZIP7 (SLC39A7)

In humans, ZIP7 (Table 4) displays a wide tissue distribution with high levels of protein expression, especially in endocrine tissues, lung, bone marrow, lymphoid tissues, and brain (notably hippocampus, cerebral cortex, basal ganglia and cerebellum), as well as in kidney and along the gastrointestinal tract, the two major sites of Zn absorption [138]. ZIP7 was first described under the name of HKE4, as the human homolog of the murine KE4 protein [209]. ZIP7 was found to be localized to intracellular membranes, such as ER [209] and Golgi apparatus, in several cell lines, including WI-38 (lung fibroblast), RWPE1 (prostate epithelium), K-562 (lymphoblast) and MCF-7 (breast cancer) [385]. In cells, ZIP7 functions as a Zn transporter involved in the homeostatic export of Zn from Golgi into the cytoplasm [209,385,386]. Expression of ZIP7 has been shown to be decreased by Zn supplementation in vitro in RWPE1 cells, consistent with its role in cellular Zn homeostasis [385].

*SLC39A7* knockdown and knockout in mice leads to a block in B cell development and decreased pre-B cell and B cell receptors signaling or embryonic lethality, respectively [210]. In the SLC39A7 knockdown mice, the remaining B-cells, exhibit reduced cytoplasmic Zn contents, similar to patients suffering from a recently discovered autosomal recessive disease caused by hypomorphic mutations of *SLC39A7*. These results are consistent with another study that also uncovered the essential role of ZIP7 in B-cells activation using an immortalized human B-cells model [387]. Moreover, Yan et al. used a morpholino-antisense ZIP7 knockdown Zebrafish model to show that depletion of the protein produces developmental defects, including a decreased head size, smaller eyes, shorter palates, and shorter and curved spinal cords, as well as a reduction in intracellular Zn content in the brain [211]. The authors further showed that the ZIP7 morpholino elicited phenotypes in Zebrafish that can be prevented by Zn supplementation [211]. Further evidence of the importance of ZIP7 in the cerebral cortex was found in a study of the impact of the induction of seizure in neonate rats, which caused a decrease in the levels of the *SLC39A7* mRNA transcript and the ZIP7 protein [388]. In this study, the change in expression of ZIP7 was correlated with neurobehavioral abnormalities they observed in the same rats, including plane righting reflex, negative geotaxis reaction reflex, and a cliff avoidance reflex. However, the same team later showed an opposite dysregulation of ZIP7 in hippocampus using a slightly different protocol of seizure induction [389]. Thus, regulation of ZIP7 in the brain is complex and its impact on the organ’s function is multilayered, highlighting the need for additional studies to further elucidate the role of ZIP7 in the CNS. Lastly, sheep and mouse models of Neuronal Ceroid Lipofuscinosis (NCL), an inherited juvenile neurodegenerative disease, showed that ZIP7 expression in brain and its cellular localization are altered. In the sheep CLN6 variant NCL model, authors detected a progressive loss of the ER/Golgi-resident Zn transporter ZIP7 in occipital lobe between 3 and 14 months of life, that can be reverted in vitro by Zn supplementation [390]. However, in the CLN6 variant mouse model of NCL, the authors measured an increase in ZIP7 expression in total brain extracts as well as in Purkinje cells, the main neuronal subpopulation of cerebellum, compared to controls [391]. In these neurodegenerative disease models, dysregulated ZIP7 expression is associated with a shorter lifespan, motor and visual decline, and abnormal Zn accumulation in the Golgi apparatus and the ER [390,391].

#### 5.2.8. ZIP8 (SLC39A8)

ZIP8 (Table 4) was first identified under the name of BIGM103 (Bacillus calmette-guerin-induced gene in monocyte clone 103) at the beginning of the 2000s [392]. ZIP8 is capable of transporting several divalent cations, including Mn, Cd, Fe, and Zn [393,394]. ZIP8 protein expression is widespread in human, with highest levels in lung, kidney [214], as well as gastrointestinal tract, the two main sites of Zn absorption [138].

One of the first identified roles for ZIP8 was its contribution to Cd-induced toxicity in testicular tissue [395], as well as in the kidney [212], where ZIP8 localizes to the apical surface of proximal tubules cells. ZIP8 has been shown to be localized to the plasma membrane of red blood cells [223]. While Zn deficiency has not been reported to affect ZIP8 localization or expression in erythrocytes, when ZIP8 was transiently overexpressed in canine polarized kidney cells (MDCK cell-line), the protein was observed to switch from the cytoplasm to apical surface of the plasma membrane in response to Zn deprivation conditions [213].

The generation of a ZIP8 hypomorphic mouse model [396] allowed investigators to demonstrate that ZIP8 is essential for development, with embryos displaying in utero reduced growth, dysmorphorgenesis, severe anemia, and developmental failure [215], with death occurring between gestational day 18.5 and 48 h postnatally [216]. These defects include a severely hypoplastic spleen, along with hypoplasia of liver, kidney, lung, and lower limbs. Moreover, mouse fetal fibroblast and liver-derived cultures showed decreased Zn cellular content, consistent with the decreased Zn levels measured in the liver and heart of the offspring [216]. These results have been confirmed by work from Dickinson and colleagues, who showed that ZIP8 homozygous KO embryos have heart morphologic defects at embryonic day 14.5 [217]. Furthermore, and consistent with its detection throughout the CNS, *SLC39A8* SNPs and mutations have been associated with specific brain structure alteration phenotypes by MRI [397,398]. Interestingly, over the past decade, *SLC39A8* polymorphisms have been largely shown to be associated with higher schizophrenia risk [399,400,401,402,403,404]. Mutations in *SLC39A8* are also associated with congenital disorder of glycosylation, type IIn [218,219], in which patients display normal localization of the mutant ZIP8 protein, but exhibit low blood Zn and Mn levels, as well as high levels of these cations in urine, suggesting renal wasting of the cations. Thus, the phenotypes associated with mutations in *SLC39A8* are complex and will require additional study.

#### 5.2.9. ZIP9 (SLC39A9)

High ZIP9 (Table 4) protein expression has been detected in most of human tissues, especially in the brain (cerebral cortex, cerebellum and hippocampus), along the gastrointestinal tract, as well as a lower expression in the kidney [138]. ZIP9 is one of the more recently discovered members of the *SLC39A* family; thus, less is known regarding its function and its localization within cells. Nevertheless, ZIP9 seems to have a *trans*-Golgi subcellular localization when overexpressed in HeLa cells, showing perinuclear and intravesicular staining by immunofluorescence as revealed by confocal laser scanning microscopy [220]. Interestingly, KO of ZIP9 in chicken lymphoblast DT40 cell line did produce any visible deleterious phenotype; no alteration in cell proliferation or cellular Zn homeostasis was observed [220]. Furthermore, these cells did not exhibit any increased sensitivity to Zn deficiency. However, generation of a CRISPR/Cas9 ZIP9 KO model in Zebrafish uncovered compelling evidence about the transporter’s importance during development. Indeed, ZIP9 deficient females had a decreased fecundity, which was associated with lower embryo viability and reduced growth of their offspring. Additionally, ZIP9 KO eggs failed to undergo chorion elevation during activation, and also exhibited decreased viability and alterations in Zn homeostasis [221].

Besides its involvement in Zn homeostasis, ZIP9 has been studied for functions not directly related to Zn. ZIP9 has been suggested to function as a membrane androgen receptor, reviewed in [405]. In addition, ZIP9 has also been investigated for its regulation of transcriptional-independent PSD-95 (Postsynaptic Density Protein 95) expression in the mouse hippocampal HT22 cell-line [406]. Therefore, ZIP9 appears to have critical roles in development and Zn homeostasis, but appears to have additional roles outside its impact on Zn transport regulation, highlighting the need for additional study of this *SLC39A* family member using mammalian animal models.

#### 5.2.10. ZIP10 (SLC39A10)

*SLC39A10* (Table 4) transcript expression is ubiquitous in humans, with highest levels detected in all the brain structures, as well as in the thyroid [138]. However, ZIP10 protein expression is quite low throughout the body, with still notable expression in both male (seminal vesicle, prostate) and female (endometrium, fallopian tube) sexual tissues, as well as in the lung, thyroid gland [138] and kidney [224]. ZIP10 is localized to the plasma membrane in the liver, brain [222], and in erythrocytes [223].

*SLC39A10* transcript expression has been shown to be induced by Zn deficiency in the gill of Zebrafish, and repressed in the gill by excess Zn [407]. The same study additionally demonstrated the Zn import function of ZIP10, similarly to what has been previously observed in both invasive (MDA-MB-231) and metastatic (MDA-MB-435S) human breast cancer cell lines [408], as well as in *Xenopus* oocytes expressing human, canine, or a *Drosophila* isoform of ZIP10 [224]. Moreover, Lichten and colleagues obtained similar results and showed that both ZIP10 mRNA and protein levels were greatly increased in mouse liver and brain, in response to Zn deficiency [222]. Using mouse erythrocytes, another study lead by Ryu and colleagues also showed an increased expression of ZIP10 in response to Zn deficiency [223]. Lastly, ZIP6 and ZIP10 have been shown to physically interact to form a heterodimer that is involved in a number of functions, including the regulation of Zn uptake in both mouse and human oocytes [409]. In addition, ZIP6 and ZIP10 have been shown to be involved in EMT (Epithelial–Mesenchymal Transition) during Zebrafish development [378]. Similar implication in EMT has also been highlighted in vitro during cancerization process in NMuMG mouse breast epithelial cell line, through the control of NCAM1 expression and its post-translational modification [410]. Another study also showed the importance of ZIP10 for mitosis triggering in breast cancer (MCF-7) and normal NMuMG cell lines [377]. Additional evidence of the function relationship between ZIP6 and ZIP10 was also uncovered by Croxford and coworkers, who showed that ZIP6 and ZIP10 were co-regulated during spermatogenesis in mouse [379].

It is, however, worth noting that, in a similar way to what we described for ZnT4, ZIP10 function may not be conserved between rodents and human. Indeed, sequences alignments of the rat version of ZIP10 (rZIP10), which is involved in Zn uptake across renal brush-border membrane, highlighted more sequence homology with human ZIP4 (hZIP4) than hZIP10 [411]. Therefore, the results of animal studies with rodents should be interpreted with caution in terms of how any experimental results may be applied to humans. Thus, additional research should be conducted to better understand the role of the ZIP10 transporter specifically in humans.

#### 5.2.11. ZIP11 (SLC39A11)

The function and regulation of *SLC39A11* (Table 4) are poorly understood. The first mention of ZIP11 in the literature was in 2001, when Bagheri-Fam and colleagues analyzed the genomic region of human SOX9 (SRY (Sex Determining Region Y)- Transcription Factor 9) and discovered an unknown gene they named *C17ORF26* [412]. Since then, ZIP11 expression was shown to be transcriptionally increased in the mouse mammary gland during lactation, where in mammary epithelial cells the transporter is localized to the Golgi [225]. Interestingly, ZIP11 has also been found localized to the nucleus of murine stomach and colon epithelial cells, where its function remains unknown [226]. In addition, another study directly demonstrated that overexpression of ZIP11 in HEK-293 cells increases cellular Zn uptake [227]. Altogether, these partial and contradictory pieces of data underscore the need of additional investigations to better understand ZIP11’s cellular function. Additionally, high levels of *SLC39A11* mRNA have been detected in mouse testis, stomach, and large intestine, with low levels of expression in the small intestine and kidney, suggesting that that ZIP11 might not be involved in Zn absorption in these tissues [227]. Nevertheless, some contradictory results were reported by Martin and coworkers, showing high ZIP11 protein expression in mouse kidney [226]. Whether expression of ZIP11 is regulated by Zn deficiency is also not clear. Yu and co-workers did not detect any variation in ZIP11 expression in stomach, testis, or liver in response to a Zn-deficient diet, but they surprisingly did detect increased expression of ZIP11 transcript and protein in spleen [227]. Conversely, Martin and colleagues demonstrated an increase in ZIP11 expression in the colon during Zn dietary restriction [226]. Thus, the control of ZIP11 expression in response to changes in Zn availability requires additional study. As of the time of this writing, no animal studies investigating the function of ZIP11 in animal models have been reported. Thus, the importance of ZIP11 for development remains unexplored.

#### 5.2.12. ZIP12 (SLC39A12)

Genomic sequences analysis by Taylor and coworkers in 2003 led to the identification of *LZT-Hs8*, which is highly expressed in brain and was later named ZIP12 [413]. Microarray results analysis further demonstrated a 47-fold higher *SLC39A12* (Table 4) expression in human brain compared to other tissues [228]. Interestingly, in vitro sequestration of extracellular Zn with TPEN caused a redistribution of the HA-tagged ZIP12 protein from the perinuclear space to the cytoplasm and plasma membrane in CHO cells [228]. The same study also demonstrated that the primary function of ZIP12 is as a Zn importer. ZIP12 also has been found to have essential role in the development of mouse nervous system. Indeed, *SLC39A12* morpholinos injected into *Xenopus* embryos produced neural tube closure defects (NTDs), including incomplete closure of the neural tube, developmental arrest after neural tube closure, or delayed closure of the neural tube, followed by lethality of all embryos before developmental stage 25 [228]. Interestingly, neuritogenesis defects in primary cultured mice neurons and Neuro-2a cells (mouse neuroblasts) transfected with ZIP12-shRNA were observed, with cells producing fewer and shorter neurites, without affecting cell viability [228]. Lastly, it is interesting to note that *SLC39A12* was recently identified as a strong candidate gene for ASD and schizophrenia susceptibility, as reviewed by Davis, underscoring the need for additional study of ZIP12’s potential pathological roles, especially in the CNS [414].

#### 5.2.13. ZIP13 (SLC39A13)

Similarly to *SLC39A12*, *SLC39A13* (Table 4) was originally discovered by sequence homology and named LZT-Hs9 [413]. ZIP13 is widely expressed in humans, and displays specifically high levels of protein expression in the esophagus, skeletal muscle, skin, and tonsils [138]. *SLC39A13* transcript expression was found to be upregulated in response to Zn deficiency in rat kidney, suggesting a potential role in Zn reabsorption during Zn deprivation [415]. When the human version of ZIP13 is overexpressed in HEK-293T cells, the protein is found localized to the Golgi apparatus and the ER [229]. Functional studies of ZIP13 overexpressed in HEK-293T cells indicate that ZIP13 mediates Zn influx and controls intracellular Zn homeostasis, demonstrating that the protein possesses Zn transport activity [229]. Loss of function mutations of *SLC39A13* have been demonstrated to cause an autosomal recessive disease called spondylocheirodysplastic form of Ehlers-Danlos syndrome (SCD-EDS) [231,232]. The main symptoms of this disease are progressive kyphoscoliosis, hypermobility of joints, hyperelasticity of skin, as well as a severe hypotonia of skeletal muscle. Consistent with the phenotype observed in humans, deletion of ZIP13 in mice produces symptomology consistent with SCD-EDS, with notable defects in osteogenesis and chondrogenesis, as well as abnormal maturation of odontoblasts [230]. Finally, KO of ZIP13 in fibroblasts causes Zn accumulation in the Golgi apparatus, providing evidence that the Zn transport function of ZIP13 is to facilitate Zn efflux from the Golgi apparatus to the cytoplasm [230]. Additional studies are needed to understand the function and regulation of ZIP13 and how alterations in its activity give rise to SCD-EDS.

#### 5.2.14. ZIP14 (SLC39A14)

ZIP14 (Table 4) has reported roles in Zn and Mn transport and was initially described as LZT-Hs4 by sequence homology analyses [413]. The transporter is expressed in the gastrointestinal tract, where it is found localized to the basolateral membrane of enterocytes [233]. ZIP14 is also expressed in the pancreas, kidney, muscle, and endocrine tissues [138], and has particularly high levels of expression in the liver [235]. Additionally, *SLC39A14* possesses three splice variants due to alternative splicing of exon 4 and 9 that have different patterns of expression: *SLC39A14* isoform 1 is ubiquitously expressed, whereas *SLC39A14* isoform 2 is not expressed in the brain, heart, skeletal muscle, or skin [236]. When ZIP14 is overexpressed in CHO cells, it localizes to the plasma membrane, where it is concentrated in cell-cell contact regions, suggesting a potential role in cell-cell adhesion [416]. In contrast, overexpression of FLAG-tagged human *SLC39A14* in the HepG2 human hepatocarcinoma cell line found ZIP14 localized to endosomes and lysosomes [234]. Similar endosomal localization for the native protein was observed in cells in the mouse duodenum and jejunum [233]. Thus, additional study of ZIP14 is needed to understand its cellular roles.

ZIP14 has significant sequence similarity with ZIP8, particularly in its the HEXXH Zn-binding motif in which the first histidine is replaced with a glutamic acid, conferring ZIP14 and ZIP8 with an ability to bind and transport metals other than Zn, including Mn [417]. Nevertheless, ZIP14 has the capacity to function as a Zn transporter, mediating Zn cellular uptake when the protein is overexpressed in CHO cells [416] as well as in HEK-293 cells overexpressing the murine version of ZIP14 [418]. Similar results were obtained in K562 human erythroleukemia cells [235] and in *Xenopus* oocytes [394] expressing the mouse version of ZIP14. The potential regulation of ZIP14 expression by Zn availability remains unclear. Indeed, induction of Zn deficiency through TPEN injections in mouse led to increased expression of *SLC39A14* mRNA in the liver [419]. More recently, it was shown that Zn supplementation of porcine oocytes causes an increase in *SLC39A14* expression [420]. Thus, regulation of ZIP14 expression by Zn availability may depend on the experimental approach and specific tissue, indicating that additional study of how ZIP14 expression is controlled is needed.

Knockout of ZIP14 in mice leads to an increase in Zn content in intestinal tissue, where it accumulates into endosomes [233]. In addition, a higher permeability of the intestinal membrane in ZIP14 KO animals was observed that was accompanied by changes in the expression of tight junction proteins. A notable decrease in PKC-ζ and claudin 1 protein expression was detected in the jejunal tissue of ZIP14 KO animals, as well as a reduced phosphorylation of occludin and an increase in claudin 2 protein expression [233]. In humans, loss of function mutations in *SLC39A14* are responsible for childhood-onset dystonia parkinsonism, with *SLC30A14* deficiency producing hypermanganesemia and neurodegeneration [236]. A similar detrimental phenotype was observed in ZIP14 KO mice, with the animals exhibiting motor deficits due to Mn homeostasis disruption, as shown by the toxic accumulation of Mn in liver and brain with Mn deficiency observed in other tissues [238]. The neurobehavioral and neurological abnormalities (reviewed in [421]) could not be rescued by Mn supplementation [238]. Loss of ZIP14 in mice also caused a defect in Mn excretion, which apparently contributed to excessive accumulation of Mn in the brain, leading to Mn neurotoxicity responsible for features of Parkinsonism [238,421].

## 6. TRPM7: A New Player in Zinc Homeostasis

Although ZIP and ZnT are the only two known families of Zn transporters in mammals, the TRPM7 channel was recently demonstrated to play a crucial role in systemic Zn homeostasis [24]. TRPM7 is a member of the transient receptor potential (TRP) ion channel family, and was originally identified as a bifunctional protein with ion channel and kinase domains [422]. The predicted molecular weight of TRPM7 is approximately 220 kDa. The channel has with a large intracellular NH_2_-terminal domain (melastatin domain) that is found in other TRPM channel family members and a large intracellular COOH-terminal domain with the serine/threonine kinase catalytic domain at its terminal end of the protein. The channel-kinase is ubiquitously expressed, with highest expression in the kidney and the intestine [422]. TRPM7 was originally shown to be required for cellular magnesium (Mg) homeostasis [423]. However, the channel, which exhibits a small inward conductance, is permeable to divalent cations other than Mg, including Ca and Zn [424]. Interestingly, the selectivity of the channel is highest for Zn [424]. Loss of TRPM7 in mouse embryonic stem cells disrupts Mg homeostasis but also causes a reduction in cellular free Zn [425]. In neurons, TRPM7 is implicated in Zn-induced toxicity during oxygen glucose deprivation, consistent with a role for the channel in influencing cellular Zn homeostasis in at least some cellular systems [426]. Whether TRPM7 has a general role in cellular Zn homeostasis still remains unknown. Mittermeier and colleagues used a TRPM7 conditional knockout mouse model to assess the role of the channel in intestinal mineral absorption [24]. When TRPM7 is deleted in the intestine, mice display a Zn-deficient phenotype, as well as developmental delay, with a high mortality of pups during pregnancy [24]. Mice lacking intestinal TRPM7 exhibit severe Zn deficiency. Serum Zn concentrations were normal in neonatal mutants at P1, but dropped precipitously to 39% of control values by P5, accompanied by a significant loss of Zn in bones. Strikingly, the mortality rate of the pups can be decreased if dams are supplemented with Zn during pregnancy. Recently, human patients with rare mutations in *Trpm7* were found in patients with hereditary hypomagnesemia with secondary hypocalcemia [427]. Patients suffered from seizures and muscle cramps due to magnesium deficiency and episodes of hypocalcemia. However, no deficiency in organismal Zn was reported, which could be due to the fact that human embryos with loss-of-function mutations in *Trpm7* do not survive to birth.

TRPM7 was the first ion channel found required for early embryonic development [428]. Studies in *Xenopus laevis* found that depletion of the channel causes gastrulation and neural tube closure defects, which could be partially suppressed by Mg supplementation or expression of the Mg transporter *SLC41A2* [429]. Unfortunately, whether Zn supplementation can also suppress defects caused by depletion of TRPM7 was never investigated. *Drosophila*, however, possesses a TRPM channel gene (*dTRPM*), which when is deleted causes a disruption of Zn homeostasis, with a Zn-deficient phenotype observed in larvae [430]. *TRPM* mutant larvae exhibited reduction in larval growth and died before the next developmental stage of pupation. The authors also observed a cellular deleterious phenotype, including defects in cell size as well as in mitochondrial morphology, which can be rescued by Zn supplementation. Although the decrease in Zn observed in *dTRPM* KO larvae tissues is less marked than what was observed in a ZnT1 knockout, as described previously [240], TRPM-mediated Zn entry remains critical for *Drosophila* development. Whether TRPM7’s developmental role in mammals involves channel-mediated Zn entry requires additional study.

## 7. Conclusions and Future Work

It is estimated that up to 17% of the world global population remains at risk from inadequate Zn intake. Zn is an essential trace mineral with an outsized role in biological systems with essential functions in nearly every cellular process. Consequently, organismal deficiency of the essential divalent cation can produce profound defects during early embryonic morphogenesis and later during brain development. Indeed, dysregulation of Zn homeostasis during brain development is extremely detrimental. Adverse events occurring during this window of susceptibility trigger molecular and signaling pathways alterations that will affect structuration of the brain, causing permanent deleterious abnormalities, including neurobehavioral impairments during childhood. These long-term consequences remain under-studied and must be further investigated.

Hence, early Zn deficiency caused by insufficient intake or inherited disorder can specifically disrupt STAT1/STAT3 [128] and Erk1/2 [119,120,121] signaling pathways as well as molecular composition of NMDA receptor in fetal brain, and NGF hippocampal contents after birth [118]. Furthermore, cellular functions are also altered in response to early Zn deficiency, with an increase in oxidative stress [128], apoptosis [81,113,119], and cytoskeleton disorganization [128,129,130]. Altogether, these events are known to cause brain developmental abnormalities, such as a reduction in neuron density [108,121], a disruption of synaptic organization, and myelination defects [131], as well as global developmental alteration of several brain structures, including cerebellum and hippocampus. While malformations and birth defects are more likely to occur under Zn deprivation conditions, long-term consequences have also been underscored, especially including neurobehavioral impairments during childhood. Indeed, incorrect brain development has direct consequences on its function, revealing alterations of motor function and cognition due to cerebellar defects, or on learning, as well as short-term and spatial memory, originating from hippocampus abnormal development.

Deficiency of Zn in adulthood also contributes significantly to health and disease, underscoring the need for stringent regulation of cellular and whole body levels of Zn. Indeed, the ion’s homeostasis in cells and in the body is controlled by a large array of transporters belonging to the ZIP and ZnT family, which can be found at the plasma membrane and localized to organelles within different cell types and organs. Moreover, an emerging paradigm is that organismal balance of Zn predominantly depends on a common gatekeeper, the channel-kinase TRPM7, that possesses an essential role during development. Similarly to ZIPs and ZnTs, TRPM7′s role and regulation must be elucidated. Although how these transporters and ion channel collectively work together still remains poorly understood, much has been learned about the function of individual Zn transporters using genetic approaches. Yet more work still remains. For example, no ZnT6 KO animal model has been reported and the importance of ZnT6 during development remains unknown. And even when KO mouse models are available, questions about the function of Zn transporters persist. Indeed, despite the discovery of the gene responsible for *AE* in human almost 20 years ago, the molecular function and regulation of ZIP4, as well as the consequences of its dysfunction, remain poorly understood [56]. Additionally, further studies are needed to understand the function and regulation of ZIP13 and how alterations in its activity give rise to SCD-EDS. An extra layer of complexity in the field is the ability of some Zn transporters to transport ions in addition to Zn. Mutations in *SLC39A8* are also associated with congenital disorder of glycosylation, type IIn [218,219], in which patients display normal localization of the mutant ZIP8 protein, but exhibit low blood Zn and Mn levels as well as high levels of these cations in urine, suggesting renal wasting of the cations. *SLC39A8* is also expressed in the brain, and *SLC39A8* SNPs and mutations have been associated with specific brain structure alteration phenotypes by MRI [397,398]; *SLC39A8* polymorphisms have also been shown to be associated with higher schizophrenia risk [399,400,401,402,403,404]. Thus, the phenotypes associated with mutations in *SLC39A8* are complex and will require additional study. In addition to its effects in pancreas, it was found that KO of ZnT8 in mice produces increased adiposity, manifestly through the control of serotonin (5-hydroxytryptamine) synthesis, underlining then a potential role of ZnT8 in control of obesity [177] and metabolic syndrome [332] and the importance of developing therapeutics targeting ZnT8 not only in diabetes. Thus, more research is needed to understand how expression of *SCL30A8* is regulated and how different mutations affect ZnT8 protein structure and function, which will be critical to make progress toward developing novel treatment strategies targeting ZnT8 [333]. Thus, research into the function and regulation of ZIP and ZnT transporters has significantly advanced our understanding of the biological importance of the trace metal and, excitingly, presented novel opportunities for the development of therapeutic interventions for many diseases.

## Figures and Tables

**Figure 1 nutrients-14-02526-f001:**
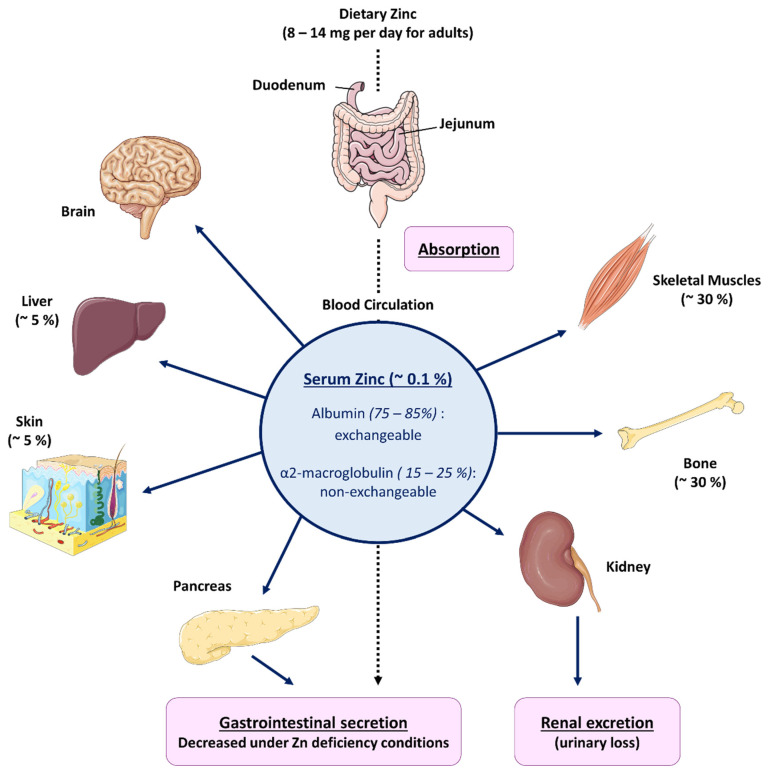
**Schematic of Zinc Distribution in the Body.** Dietary zinc is absorbed (uptake) in the small intestine (duodenum and jejunum) and about 60% and 30% will be stored in the bone and skeletal muscles, respectively. Liver and skin both represent about 5% of zinc storage in the body. The remaining is transported to other organs, such as the brain, the pancreas, the kidney, or the mammary gland through blood circulation, constituting only about 0.1% of the total zinc of the body. In the blood, zinc is found under two forms: bound to albumin (about 75 to 85% of the serum zinc), constituting the exchangeable pool of zinc; or bound to 2-macroglobulin (15 to 25% of serum zinc), which is the non-exchangeable reserve of zinc. Excess of zinc is mainly excreted through gastrointestinal excretion, depending on the zinc status; or by renal excretion, that represents a minor and less regulated Zn egestion through urinary loss. These complex mechanisms of absorption, transport, excretion, and reabsorption are tightly controlled by two families of Zn transporters: the ZnTs and the ZIPs proteins.

**Figure 2 nutrients-14-02526-f002:**
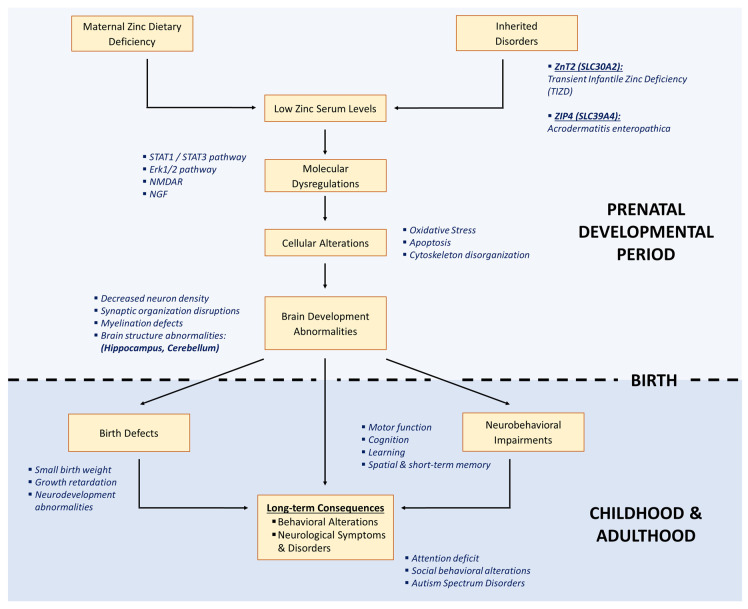
**Long-term consequences of zinc deprivation during brain development.** Dietary zinc is absorbed (uptake) in the small intestine (duodenum and jejunum) and about 60% and 30% will be stored in the bone and skeletal muscles, respectively. Liver and skin both represent about 5% of zinc storage in the body. The remaining is transported to other organs, such as the brain, the pancreas, the kidney, or the mammary gland through blood circulation, constituting only about 0.1% of the total zinc of the body. In the blood, zinc is found under two forms: bound to albumin (about 75 to 85% of the serum zinc), constituting the exchangeable pool of zinc; or bound to 2-macroglobulin (15 to 25% of serum zinc), which is the non-exchangeable reserve of zinc. Excess of zinc is mainly excreted through gastrointestinal excretion, depending on the zinc status, or by renal excretion, which represents a minor and less regulated zinc egestion through urinary loss. These complex mechanisms of absorption, transport, excretion, and reabsorption are tightly controlled by two families of zinc transporters: the ZnTs and the ZIPs proteins. Early zinc deficiency caused by dietary deprivation or inherited mutation of ZnT2 (Transient Infantile Zinc Deficiency, TIZD) or ZIP4 (*Acrodermatitis enteropathica*) during prenatal brain development is the source of molecular dysregulations that can then cause cellular alteration and consequential structural and functional brain development abnormalities. Altogether, these deleterious events can be the source of birth defects and/or neurobehavioral impairments that will have mild to severe adverse long-term consequences during childhood and adulthood. Abbreviations: ZnT: Zinc Transporter; SLC30A2: Solute Carrier family 30 member 2; ZIP4: Zinc-regulated, Iron-regulated transporter-like Protein; SLC39A4: Solute Carrier family 39 member 4; STAT1/3: Signal Transducer and Activator of Transcription 1/3; Erk1/2: Extracellular signal-Regulated Kinases 1/2; NMDAR: N-Methyl-D-Aspartate Receptor; NGF: Nerve Growth Factor.

**Table 1 nutrients-14-02526-t001:** Dietary Reference Intakes for Zinc.

	Age	Male	Female	Pregnancy	Lactation
Infants	0–6 months	2 mg *	2 mg *		
	7–12 months	3 mg	3 mg		
Children	1–3 years	3 mg	3 mg		
	4–8 years	5 mg	5 mg		
	9–13 years	8 mg	8 mg		
	14–18 years	11 mg	9 mg	13 mg	14 mg
	≥18 years	11 mg	8 mg	11 mg	12 mg

This table represents the Recommended Dietary Allowances (RDAs), according to the Dietary Reference Intakes (DRIs) developed by the Food and Nutrition Board (FNB) at Institute of Medicine of the National Academies. RDA is described as the average daily zinc intake that is set to meet the needs of almost (97 to 98%) all individuals of a group. When evidence was insufficient to determine an RDA, Adequate Intakes (AI) have been developed. * Thus, AI is the mean of zinc intake that is assumed to ensure nutritional adequacy for breastfed infants.

**Table 2 nutrients-14-02526-t002:** Zinc Content and Daily Value of Selected Food.

Food	Weight (g)	Measure	Milligrams (mg) Per Serving	Percent DV *
Oysters, eastern, wild, raw	84	6.0 medium	33	300
Beef chuck, blade roast, separable lean and fat, trimmed to 0” fat, all grades, cooked, braised	235	1.0 piece, cooked, excluding refuse (yield from 1 lb raw meat with refuse)	20.16	183
Soybeans, mature seeds, raw	186	1.0 cup	9.1	83
Lamb, domestic, rib, separable lean only, trimmed to 1/4” fat, choice, cooked, broiled	147	1.0 piece, cooked, excluding refuse (yield from 1 lb raw meat with refuse)	7.75	70
Nuts, cashew nuts, dry roasted, with salt added	137	1.0 cup, halves and whole	7.67	70
Peas, green, split, mature seeds, raw	197	1.0 cup	6.99	63
Oats	156	1.0 cup	6.19	56
Crustaceans, spiny lobster, mixed species, cooked, moist heat	85	3.0 oz	6.18	56
Cocoa dry powder, unsweetened	86	1.0 cup	5.86	53
Desserts, mousse, chocolate, prepared-from-recipe	808	1.0 recipe yield	5.17	47
Cheese, pasteurized process, swiss	140	1.0 cup	5.05	46
Pork, fresh, shoulder, whole, separable lean and fat, cooked, roasted	135	1.0 cup	5.01	46
Chicken, broilers or fryers, dark meat, drumstick, meat only, cooked, roasted	96	1.0 drumstick without skin	2.46	12
Pepperoni, beef and pork, sliced	85	3.0 oz	2.07	19
Yogurt, vanilla, non-fat	245	8 fl oz	2.03	18
Fast foods, taco with beef, cheese and lettuce, soft	102	1.0 each taco	1.4	13
Fish, tuna, light, canned in oil, drained solids	146	1.0 cup	1.31	12
Milk, reduced fat, fluid, 2% milkfat, with added nonfat milk solids and vitamin A and vitamin D	245	1.0 cup	0.98	9
Bananas, raw	225	1.0 cup	0.34	3
Lettuce, butterhead (includes boston and bibb types), raw	55	1.0 cup	0.11	1
Cabbage, common (danish, domestic, and pointed types), stored, raw	35	0.5 cup	0.06	0.5

Zinc content of a selection of food sorted by percent Daily Value (DV) *. The FDA reported that the DV index for zinc is 11 mg for adults and children aged 4 years and older. According to the FDA, a source of food providing 20% of the DV or more is considered as high source of a nutrient. The food selection as well as their zinc content have been selected from the National Nutrient Database for Standard Reference, Release 28 (Slightly revised) FoodData Central (FDC) of the U.S. Department of Agriculture (USDA). Besides supplementation, patients suffering from Zn deficiency are advised to eat more food considered as rich in zinc, including red meat, poultry, beans, seeds, and nuts. On the other hand, vegetables and fruits are overall considered as having lower Zn contents, making it challenging for vegetarians and vegans.

**Table 3 nutrients-14-02526-t003:** Mammal ZnT proteins, phenotypes of KO or mutant animals and human inherited disorders associated with their mutations.

Protein	Gene	Subcellular Localization	Tissue Expression	Phenotypes of KO or Mutant Animals	Human Inherited Diseases Associated with Mutations
ZnT1	*SLC30A1*	Plasma membrane (basal pole of enterocytes) [137]	Ubiquitous [138]	KO: embryonic lethal [139] Heterozygous KO: Higher sensitivity to maternal Zn deficiency [139]	
ZnT2	*SLC30A2*	Endosome and lysosome [140]	Human: pancreas [141], mammary gland [142], prostate [143], kidney [144], small intestine [138] Rat: enterocytes [145]	KO: Electrophysiological abnormalities of mossy fiber synapses onto CA3 pyramidal neurons [146]	Transient Infantile Zinc Deficiency (TIZD) [147]
ZnT3	*SLC30A3*	Synaptic vesicles [148,149]	Brain [150]: synaptic vesicles of glutamatergic neurons of hippocampus [151,152] and cerebral cortex [151,152], testis [150], epididymis [138]	KO: Fear memory decrease [153] Long-term memory impairment in males [154] Spatial navigation and memory defects [154,155] Autism-like phenotype in males: decreased social interaction, decreased locomotion and increased anxiety-like behavior [156]	
ZnT4	*SLC30A4*	Plasma membrane, intracellular vesicles [144,157]	Human: wide expression, enriched in brain (hippocampus and caudate), thyroid, lung, testis, heart, skin, pancreas [138] Mouse: mammary gland [158,159] Human and mouse: *SLC30A4* transcript expression in small intestine but not protein [138,160]	Mutations: low zinc in breast milk (lethal milk mouse) [158,159,161]	
ZnT5	*SLC30A5*	Golgi [162], plasma membrane [163,164], cytoplasmic vesicles [165]	Ubiquitous [138] with higher levels in stomach (parietal cells), duodenum and jejunum [166]	Homozygous loss of function: perinatal death, cardiomyopathy, hydrops fetalis, cystic hygroma [167] KO: growth defects, osteopenia, muscle weakness and male-specific sudden cardiac death [168]	
ZnT6	*SLC30A6*	Golgi, intracellular vesicles [169]	Human: ubiquitous [138] Mouse: high levels in brain, lung and small intestine [169] (duodenum and jejunum) [166]	-	
ZnT7	*SLC30A7*	Golgi, intracellular vesicles [170]	Human: ubiquitous with higher levels in small intestine and colon, but low levels in the brain [138] Mouse: *SLC30A7* transcript expression in liver, kidney, spleen and small intestine [170] but protein only detected in duodenum and jejunum (high) and along the gastrointestinal tract (lower expression) [166]	KO: Irreversible growth retardation, small weight, zinc deficiency [171]	
ZnT8	*SLC30A8*	Secretory granules [172,173]	Human: pancreas [172] Mouse: pancreatic beta and alpha islet cells, thyroid, adrenal gland [173]	Pancreatic function defects [172,173,174,175,176] Increased adiposity [177]	
ZnT9	*SLC30A9*	Cytoplasm, nucleus [178,179,180]; endoplasmic reticulum [181]	Ubiquitous [138] with higher levels in cerebellum, skeletal muscles, thymus and kidney [181]	No KO model, but (c.1047_1049delGCA, p.(A350del)) mutation causes neurological alterations and severe intellectual disability [181]	
ZnT10	*SLC30A10*	Golgi, plasma membrane [182]	Small intestine, liver, cerebral cortex [138,182], and retina [181]	No changes in plasma or tissue Zn concentration, but high manganese contents in plasma, brain and liver [183]	Juvenile-onset dystonia, adult-onset parkinsonism, severe hypermanganesemia, polycythemia, and chronic hepatic disease (steatosis and cirrhosis) [184,185]

**Table 4 nutrients-14-02526-t004:** Mammal ZIP proteins, phenotypes of KO or mutant animals and human inherited disorders associated with their mutations.

Protein	Gene	Subcellular Localization	Tissue Expression	Phenotypes of KO or Mutant Animals	Human Inherited Diseases Associated with Mutations
ZIP1	*SLC39A1*	Plasma membrane [186], endoplasmic reticulum [186], intracellular vesicles [187,188]	Ubiquitous [138] with higher levels in small intestine human [186] and brain (hippocampus, thalamus and cerebral cortex) [189]	KO: higher sensitivity to zinc deficiency causing developmental malformations and increased lethality [190,191,192]	
ZIP2	*SLC39A2*		Wide expression: brain (cortex, cerebellum, basal ganglia), endocrine tissues, lungs, gastrointestinal tract, liver, pancreas, kidney, heart [138]	KO: higher sensitivity to zinc deficiency causing developmental malformations [193]	
ZIP3	*SLC39A3*	Golgi (zinc sufficient conditions), plasma membrane (zinc depletion) [194]	Wide expression with higher levels along gastrointestinal tract [138]	KO: higher sensitivity to zinc deficiency causing developmental malformations [195] ZIP1-ZIP3 double-KO: high sensitivity to zinc deficiency and higher rates of developmental malformations than ZIP1 single KO [190] ZIP1-ZIP2-ZIP3 triple-KO: higher sensitivity to zinc deficiency and 80% of developmental malformation (with 60% of severe defects) [191]	
ZIP4	*SLC39A4*	Intracellular vesicles (zinc sufficient conditions), plasma membrane (zinc depletion) [196,197,198]	Human: duodenum, jejunum [23,56,138] (Enterocytes [197,198,199,200]), pancreatic β cells Rat: in vitro cultures of astrocytes and neurons [201]	Homozygous KO: early embryonic lethal [202] Heterozygous KO: developmental defects (severe growth retardation, exencephaly), higher sensitivity to zinc deficiency [202] Intestinal epithelium conditional KO: intestinal stem-cell niche disruption; epithelium disorganization [203]	*Acrodermatitis enteropathica (AE)*: “genetic zinc deficiency” [23,54,55,56]
ZIP5	*SLC39A5*	Plasma membrane (basolateral pole of polarized cells) [200,204], intracellular vesicles [200,205]	Liver, pancreas [204], peripheral blood mononuclear cells [206]	KO: zinc accumulation in liver and pancreas [207]	
ZIP6	*SLC39A6*	Plasma membrane [208]	Ubiquitous with higher levels in cerebellum, adrenal gland, endometrium [138]	-	
ZIP7	*SLC39A7*	Golgi, endoplasmic reticulum [209]	Wide expression with higher levels in endocrine tissues, lung, bone marrow, lymphoid tissues, brain (hippocampus, cerebral cortex, basal ganglia, cerebellum), kidney, gastrointestinal tract [138]	Hypomorphic: developmental defects, B-cell maturation arrest [210] KO: embryonic lethal [210] KD (Zebrafish): developmental defects, reduction in brain zinc contents [211]	Hypomorphic mutations of SLC39A7 (autosomal recessive): absent B cells, agammaglobulinemia, early onset infections [210]
ZIP8	*SLC39A8*	Plasma membrane (apical pole of polarized cells) [212], cytoplasm [213]	Wide expression with higher levels in lung, kidney [214], gastrointestinal tract [138]	Hypomorphic: growth retardation, developmental failure [215], late developmental death [216] KO: heart morphology defects [217]	Congenital disorder of glycosylation, type IIn (autosomal recessive): intellectual disability, cerebellar atrophy, low blood zinc and manganese, zinc and manganese renal wasting [218,219]
ZIP9	*SLC39A9*	Golgi, intracellular vesicles [220]	Wide expression with higher levels in brain (cortex, cerebellum, hippocampus), gastrointestinal tract, kidney [138]	KO (Zebrafish): developmental abnormalities, decreased viability, zinc homeostasis disruption [221]	Autosomal recessive cerebro-renal symptom: early neurological deterioration, severe intellectual disability, ataxia, camptocormia, early-onset nephropathy, hypertension [181]
ZIP10	*SLC39A10*	Golgi [182], plasma membrane [182,222,223]	Male (seminal vesicle, prostate) and female (endometrium, fallopian tube) sexual tissues, lung, thyroid [138], kidney [224]	-	
ZIP11	*SLC39A11*	Golgi [225], nucleus [226]	Mammary gland [225], colon [226], spleen [227]	-	
ZIP12	*SLC39A12*	Golgi (zinc sufficient conditions), plasma membrane (zinc deficient conditions) [228]	Brain [228]	KO (Zebrafish): neural tube closure defects, reduced viability [228]	
ZIP13	*SLC39A13*	Golgi, endoplasmic reticulum [229]	Wide expression, with higher levels in esophagus, skeletal muscle, skin, tonsil [138]	KO: osteogenesis and chondrogenesis defects, abnormal maturation of odontoblasts and fibroblasts [230]	Spondylocheirodysplastic form of Ehlers-Danlos syndrome (autosomal recessive): progressive kyphoscoliosis, hypermobility of joints, hyperelasticity of skin, severe hypotonia of skeletal muscles [231,232]
ZIP14	*SLC39A14*	Plasma membrane (basolateral pole of enterocytes) [233], endosome, lysosome [233,234]	Wide expression: digestive and gastrointestinal tract [233], pancreas, kidney, muscles, endocrine tissues [138] with highest levels in liver [235] Isoform 1: ubiquitous [236] Isoform 2: not expressed in brain, heart, skeletal muscle, and skin [236]	KO: Growth retardation, impaired gluconeogenesis [237] Intestinal zinc accumulation in endosomes, higher permeability of intestinal membrane, dysregulation of tight junction protein expression [233] Manganese excretion defect, liver and brain (neurotoxicity) manganese accumulation, manganese deficit in other tissues, motor deficit [238]	Childhood-onset dystonia parkinsonism: hypermanganesemia, neurodegeneration [236]

## Data Availability

Not applicable.

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
