# Peer review of "Impact of Zinc Transport Mechanisms on Embryonic and Brain Development"

_nutrients, 2022, doi:10.3390/nu14122526_

Round 1

Reviewer 1 Report

This review is well put together and quite extensive in its nature. It is generally well written and I enjoyed reading it. I mention some small changes needed below

Table 1, my copy has two table 1 so one needs removing. It states less than 18 years and should be more than 18 years

line 270-273, two zinc transporters are mentioned but no introduction has been given at this stage to these zinc transporters and there is no suggested consequence 

line 442-446, it mentions changes in STAT proteins but does not explain what this means of what effect it should have or why.

line 247, lover should be lower

line 488-489, this is not clear, ZnT transporters do not all transport out of the cell as implied, in fact just ZnT1 does, they transport into compartments and out of the cytoplasm. Likewise not all ZIP transporters are on the plasma membrane and therefore unable to transport zinc or other ions into cells by that route.

line 495, I am unconvinced about the value of including TRPM7 in this review and if it remains perhaps it should be included in a table as well. Perhaps some additional information of structure may be beneficial to make more of a comparison to the zinc transporters.

line 1215, this is not a sentence.

table 3 and 4, I find it untidy that some zinc transporter names are on two lines because the column is too narrow.

Author Response

  • Point1: Table 1, my copy has two table 1 so one needs removing. It states less than 18 years and should be more than 18 years

Thank you for catching our error. We corrected this issue and removed the extra table. Also, we replaced “less than” with “more than”.

  • Point 2: Line 270-273, two zinc transporters are mentioned but no introduction has been given at this stage to these zinc transporters and there is no suggested consequence

Thank you for highlighting the lack of context for these two transporters. Because we prefer to properly introduce them later, and also because this piece of information was not crucial, we decided to delete the whole sentence.

  • Point 3: Line 442-446, it mentions changes in STAT proteins but does not explain what this means of what effect it should have or why.

Thank you for highlighting it. We changed it to make it more detailed and explain more about STATs importance (see below):

STAT1 and STAT3 transcription factors are known to play major roles in CNS development through their role in cell proliferation, cell death, survival and differentiation [127–130]. Using Zn depleted IMR-32 neuroblastoma cells, authors also showed a decrease in STAT1 and STAT3-dependent gene transactivation as well as an increase in oxidative stress markers and abnormalities in cytoskeleton dynamics in both IMR-32 cells and fetal brains, demonstrating that changes in the cytoskeleton are also observed in response to Zn deficiency [131].

  • Point 4: Line 247, lover should be lower.

Corrected

  • Point 5: Line 488-489, this is not clear, ZnT transporters do not all transport out of the cell as implied, in fact just ZnT1 does, they transport into compartments and out of the cytoplasm. Likewise not all ZIP transporters are on the plasma membrane and therefore unable to transport zinc or other ions into cells by that route.

Thank you for noticing this incoherence. The idea here is to give a quick introduction of ZnTs and ZIPs transporters. In order to not make it too complicated, we replaced the sentence with the following:

ZnTs and ZIPs proteins display distinct and specific patterns of expression within the cell, allowing the transport of Zn ions between the different cellular compartments and therefore participating in cellular Zn homeostasis.

  • Point 6: Line 495, I am unconvinced about the value of including TRPM7 in this review and if it remains perhaps it should be included in a table as well. Perhaps some additional information of structure may be beneficial to make more of a comparison to the zinc transporters.

We thank the reviewer for offering this perspective. Our decision to include TRPM7 in the review is to introduce the channel to researchers in the Zinc transporter field that may not be familiar with the function of the channel and its newfound connection to intestinal Zn absorption. TRPM7 has been implicated in both cellular and systemic Zn homeostasis, we thought it prudent to include the manuscript with the intention of making it more comprehensive. Rather than incorporate TRPM7 into the Table devoted to Zn transporters, we chose instead to keep a description of TRPM7's properties contained with the section dedicated to the channel, since it didn't make sense to make a separate table for just TRPM7. We did, however, incorporate the reviewer's suggestion to include information about the structure of the channel as well as its properties, which I think will be helpful to those scientists unfamiliar with TRPM7.

  • Point 7: Line 1215, this is not a sentence.

Thank you for noticing. We modified the sentence as suggested:

Whether TRPM7 has a general role for TRPM7 in cellular Zn2+ homeostasis still remains unknown

  • Point 8: Table 3 and 4, I find it untidy that some zinc transporter names are on two lines because the column is too narrow.

We modified the columns to make them fit better with the tables and avoid having the transporters names on two lines.

Reviewer 2 Report

This is a very comprehensive review article with all up-to-date information in the field. Very well-written and easy to follow. There are barely any mistakes found. It will surely provide the most detailed information for researchers in the field. 

Author Response

This is a very comprehensive review article with all up-to-date information in the field. Very well-written and easy to follow. There are barely any mistakes found. It will surely provide the most detailed information for researchers in the field. 

We thank reviewer 2 for their supportive comments and are glad they enjoyed reading our manuscript.